# MULTI-OBJECTIVE MULTI-SOLUTION TRANSPORT

## ABSTRACT

In the realm of multi-objective optimization, we introduce "**M**ulti-**o**bjective multi-**s**olution **T**ransport (MosT)", a novel solution for optimizing multiple objectives that employs multiple solutions. The essence lies in achieving diverse trade-offs among objectives, where each solution performs as a domain expert, focusing on specific objectives while collectively covering all of them. Traditional methods often struggle, especially when the number of objectives greatly outnumbers the number of solutions, leading to either subpar solutions or objectives that have been essentially ignored. MosT addresses this by formulating the problem as a bi-level optimization of weighted objectives, where the weights are defined by an optimal transport between the objectives and solutions. Our newly developed algorithm not only ensures theoretical convergence to various Pareto front solutions but is also adaptive to cases where objectives outnumber solutions. We further enhance its efficiency by introducing a solution-specialization curriculum. With proven applications in federated learning, fairness-accuracy trade-offs, and standard MOO benchmarks, MosT distinctly outperforms existing methods, delivering high-quality, diverse solutions that profile the entire Pareto frontier, thus ensuring balanced trade-offs across objectives.

## 1 INTRODUCTION

The underlying goal of many machine learning problems is to simultaneously optimize multiple objectives. Usually, however, there does not exist one solution (or model) that is optimal for all the objectives at the same time. Optimizing a naïve linearization (i.e., linearly combining all objectives into one) may collapse to trivial solutions for single objectives and may lead to poor performance. Instead, multi-objective optimization (MOO) aims to find a solution on the Pareto frontier where no objective can be improved without degrading others. In MGDA (Désidéri, 2012), for example, this is achieved by finding a common direction to update the model along which no objective degrades. Unlike naïve linearization, MOO has the potential to visit different regions of the Pareto frontier, which can provide diverse trade-offs between objectives. However, the trade-off ratio of MOO solutions is not fully controllable even using reference vectors to guide the search in the objective space like EPO (Mahapatra & Rajan, 2020), especially when the Pareto frontier is unknown and complicated (e.g., non-smooth or discontinuous).

Moreover, most MOO approaches focus on two or three objectives but hardly scale up to many objectives. As objectives increase, it is less plausible that they will reach an agreement on a single solution. Instead of balancing all of them, it is more appealing to find multiple diverse yet complementary solutions on the Pareto frontier each focusing on a local domain of objectives. This problem of finding $m$ Pareto solutions (or training $m$ models) for $n$ objectives can be understood as a multi-solution extension of MOO or a mixture of experts (MoE) (Jacobs et al., 1991) for multiple objectives. However, as $n$ increases, the reference vectors become high-dimensional and a uniform exploration of the Pareto frontier is computationally prohibitive and thus practically infeasible. Therefore, it is challenging to pre-determine the search regions of a few representative solutions profiling the Pareto frontier.

In this paper, we ask: *can we leverage the MOO structure to guide the exploration of $m$ solutions on the high-dimensional (big-$n$) Pareto frontier?* For example, some objectives are consistent and share similar goals so optimizing them by the same model can bring common improvement, while optimizing separate models for objectives with mutual conflicts can effectively avoid poor performance and the tug-of-war among them. Hence, a matching between models and objectives after every MOO step is able to explore the correlation among $n$ objectives along the optimization

trajectory. Specifically, the matching relations can be denoted by a weight matrix $\Gamma \in \mathbb{R}_+^{n \times m}$ where $\Gamma_{\cdot,j}$ reweighs the $n$ objectives that model-$j$ aims to optimize, steering its optimization (MOO) towards a domain expert focusing on a locally consistent subset of objectives. With different but complementary (i.e., every objective is equally covered) and balanced (i.e., no model is dominating on most objectives) $\Gamma_{\cdot,j}$ for the $m$ models that adjust their MOO directions per step, it helps find $m$ diverse yet complementary solutions on the Pareto frontier.

In this paper, we mainly investigate a more challenging and under-explored class of the above problem when the number of objectives $n$ is much larger than the number of solutions $m$, i.e., $n \gg m$. This is emerging in a variety of machine learning problems that involve many (i.e., big-$n$) users, domains, or evaluation criteria, each associated with a different training objective, but the total available data or computation can only support the training of $m \ll n$ models. To this end, we formulate the matching as an *optimal transport* (OT) between the $m$ models and $n$ objectives, as the two marginal constraints $\Gamma \mathbf{1}_m = \alpha$[1] and $\Gamma^\top \mathbf{1}_n = \beta$ in OT allow us to control the ratio of $n$ objectives assigned to each model and the ratio of $m$ solutions optimized for each objective. For example, with uniform marginal $\alpha = (1/n)\mathbf{1}_n$, the $m \ll n$ models are enforced by OT weights $\Gamma$ to focus on different but complementary subsets of objectives[2]. Analogously, with uniform $\beta = (1/m)\mathbf{1}_m$, the training loads for the $m$ models are balanced so no one will dominate the others on a majority of objectives.

We propose an intuitive algorithm for the above "*Multi-objective multi-solution Transport* (**MosT**)" problem. It can be formulated as a bi-level MOO, where the upper-level is the $\Gamma$-reweighted MOO problems for the $m$ models and the lower-level constraint is the OT optimizing $\Gamma$ (objective-solution matching). The algorithm of MosT theoretically converges to $m$ solutions on the Pareto frontier by alternating between MOO and OT, which have been implemented in off-the-shelf solvers. We further extend MosT to the $n \ll m$ case by augmenting the $n$ objectives with random linear combinations of them ("**MosT-E**"). In addition, we introduce a curriculum for better model specialization by gradually varying the marginals $\alpha$ and $\beta$ in MosT, which gives higher priority to models selecting objectives at the earlier stages and then transits to a higher priority of objectives selecting the best models.

We apply MosT to several fundamental machine learning problems including federated learning (FL) (McMahan et al., 2017), fairness-accuracy trade-offs, and other MOO benchmarks, which cover both the $n \gg m$ case (e.g., $n$ local clients and $m$ global models in FL) and the $n \ll m$ case (e.g., $n = 2$ in fairness-accuracy trade-offs). MosT consistently outperforms MOO baselines on different metrics for those tasks and exhibits promising advantages in finding diverse higher-quality solutions better profiling the Pareto frontier.

## 2 RELATED WORK

**Single-Solution Multi-Objective Optimization (MOO).** The classic goal of MOO is to find some solution that lies on the Pareto front of multiple objectives (Désidéri, 2012; Roy et al., 2023; Halffmann et al., 2022; Miettinen, 1999). One approach to achieving this is to solve a linearized aggregation (i.e., weighted average) of all objectives. However, linearization, despite being formulated under a broad coverage of objective weights, may always result in solutions distributed a small area on the entire Pareto front (Boyd & Vandenberghe, 2004). The multiple-gradient descent algorithm (MGDA) (Désidéri, 2012) is widely used due to its capability to handle complicated Pareto fronts and compatibility with gradient-based optimization. In this work, we use MGDA-style optimization methods in our algorithm, and compare with linearization-based objectives (with different weights) empirically (Section 5), showing that our approach can find more diverse Pareto stationary solutions.

**Multi-Solution MOO.** One line of work that aims to discover diverse solutions across the entire Pareto front builds upon MGDA and guides the search process via constraints of preference vectors (Lin et al., 2019; Mahapatra & Rajan, 2020) or constraints of other objectives (Zafar et al., 2017). These methods does not generalize well to the setting where there are many objectives (constraints) or the model dimension is large, since the number of preference vectors to explore the whole Pareto front may depend exponentially on these factors in the worst scenario (Emmerich & Deutz, 2018). Even in the setting where there are only a few (e.g., two or three) objectives, diversity of the preference vectors in the action space (during exploration) may not translate to diversity in the solution

---

[1] $\mathbf{1}_m$ is the $m$-dimensional all-one vector and $\Gamma \mathbf{1}_m$ computes the row-wise sum of $\Gamma$.

[2] Otherwise, there will be uncovered objectives and $\Gamma$ will violate constraint $\Gamma \mathbf{1}_m = \alpha = (1/n)\mathbf{1}_n$.

space. We showcase the superiority of MosT relative to these works in Section 5. Some works that balance between Pareto optimality and solution diversity cannot guarantee the final solutions are on the Pareto front (Liu et al., 2021b). For gradient-free methods, evolution strategies or Bayesian optimization (Coello, 2006; Sindhya et al., 2012) has been explored to find multiple (as opposed to one) Pareto stationary solutions. However, they are usually not efficient when solving practical MOO problems in machine learning due to the lack of gradient information (Liu et al., 2021a; Momma et al., 2022); hence, we do not compare with those methods.

**Applications in Machine Learning.**    Similar as prior works (e.g., Zitzler et al., 2000; Mahapatra & Rajan, 2020), we apply MosT to a toy problem and addressing fairness-utility trade-offs (two objectives). Additionally, we demonstrate the effectiveness of MosT on a larger-scale task of cross-device federated learning (McMahan et al., 2017). There is also extensive prior research on personalized federated learning (e.g., Smith et al., 2017; Ghosh et al., 2020; Wu et al., 2022), i.e., outputting multiple related models, instead of one, to serve all clients. Our approach can be viewed as a personalization objective in this context. Note that our goal is not to achieve the highest average accuracy for federated learning, but rather, *explicitly* balance multiple objectives and guarantee that all output solutions are Pareto stationary.

# 3    MoST: MULTI-OBJECTIVE MULTI-SOLUTION TRANSPORT

Let $L_i(\cdot)$ $(i \in [n])$ denote the empirical loss function of the $i$-th objective. When the number of objectives $n$ is much larger than the number of solutions $m$, it is possible that the learnt solutions (for example, simply by running MGDA for $m$ times with different randomness) cannot cover representative regions on the Pareto front. To address this, we use a cost matrix $\Gamma \in \mathbb{R}_+^{n \times m}$ with constraints $\Gamma \mathbf{1}_m = \alpha$ and $\Gamma^\top \mathbf{1}_n = \beta$ on top of the losses to enforce a balanced matching between objectives and solutions. Our multi-objective multi-solution transport (MosT) objective is as follows:

Find $\{\theta_{1:m}\}$ such that every $\theta_j$ $(j \in [m])$ is the Pareto solution of $m$ weighted objectives
$$\min_{\theta_j} (\Gamma_{1,j} L_1(\theta_j), \cdots, \Gamma_{n,j} L_n(\theta_j)), \text{ where} \tag{1}$$
$$\Gamma \in \min_{\Gamma \in \Omega} \sum_{i=1}^n \sum_{j=1}^m \Gamma_{i,j} L_i(\theta_j), \ \Omega \triangleq \{\Gamma \in \mathbb{R}_+^{n \times m} : \Gamma \mathbf{1}_m = \alpha, \Gamma^\top \mathbf{1}_n = \beta\}. \tag{2}$$

$\alpha \in \Delta^n$, $\beta \in \Delta^m$ are two tunable vectors on $n$- and $m$-dimensional probability simplexes, respectively. We encourage diversity and nondegeneracy of solutions by setting these vectors to follow uniform distributions, i.e., $\alpha = \mathbf{1}_n/n, \beta = \mathbf{1}_m/m$. As such, the constraint set $\Omega$ prevents the degenerating case where all objectives are served by a small subset of the solutions. By definition, any Pareto solution of the rescaled objectives in Eq. (1) is also a Pareto solution of the original objectives $(L_1, \cdots, L_n)$.

At a high level, the bi-level optimization problem described above can be decoupled into two sub-problems (over $\Gamma$ and $\theta_{1:m}$) when fixing one variable and optimizing the other. At each outer iteration, we first solve (2) completely by running an off-the-shelf optimal transport (OT) solver (e.g., IPOT (Xie et al., 2020)). Then we optimize (1) by running a variant of vanilla MGDA with a min-norm solver (Désidéri, 2012). The exact algorithm is summarized in Algorithm 1 below.

---

**Algorithm 1** Multi-Objective Multi-Solution Transport (MosT)

---
1: **Input:** $n$ objectives $\{L_i(\cdot)\}_{i=1}^n, \alpha \in \Delta^n, \beta \in \Delta^m, \eta, K, \epsilon > 0$
2: **Initialize:** $m$ solutions $\theta_{1:m}$
3: **for** $t \in \{1, \cdots, T\}$ **do**
4:     $\Gamma^t \leftarrow$ solution of Eq. (3) by an optimal transport (OT) solver given $\theta_{1:m}^t$; $\Gamma^t \leftarrow \Gamma^t + \epsilon I$
5:     **for** $j \in \{1, \cdots, m\}$ **do**
6:         **for** $k \in \{1, \cdots, K\}$ **do**
7:             $d_j \leftarrow$ Eq. (6), where $\lambda^*$ is achieved by a min-norm solver for Eq. (5) given $\Gamma^t$ and $\theta_j$.
8:             $\theta_j \leftarrow \theta_j + \eta d_j$ ;
9:         **end for**
10:        $d_j^t \leftarrow d_j; \theta_j^t \leftarrow \theta_j$
11:    **end for**
12: **end for**
13: **return** $\theta_{1:m}^T$

---

From Algorithm 1, in each iteration, we first optimize $\Gamma^t$ with $\theta_{1:m}^t$ fixed, i.e., finding the optimal transport (or matching) between the $n$ objectives and the $m$ models by solving the following optimal transport problem with existing algorithms (Xie et al., 2020):

$$\min_{\Gamma \in \Omega} \sum_{j=1}^m \sum_{i=1}^n \Gamma_{i,j} L_i(\theta_j). \tag{3}$$

We then add a small constant $\epsilon$ to every cell of $\Gamma^t$. In our experiments, the $\epsilon$ is a tunable hyperparameter as detailed in Section 5.1. Fixing the optimal $\Gamma$, we then optimize for a reweighted version of MOO across objectives $(\Gamma_{1,j} L_1(\theta_j), \cdots, \Gamma_{n,j} L_n(\theta_j))$ for each solution $\theta_j$, $j \in [m]$. To find Pareto stationary solutions, similar as MGDA, we aim to find the common-descent directions $d_{1:m}$ for $\theta_{1:m}$ to guarantee that all objective values will decrease (or at least not increase) at each iteration. This reduces to solving $m$ MGDA-type MOO problems (more background in Appendix A) in parallel, i.e., for every solution $j \in [m]$, we aim to find $d_j$ by solving

$$\min_{d_j} \max_{i \in [n]} d_j^\top \Gamma_{i,j} \nabla_{\theta_j} L_i(\theta_j) + \tfrac{1}{2} \|d_j\|_2^2, \tag{4}$$

which rescales each objective's gradient $\nabla_{\theta_j} L_i(\theta_j)$ by $\Gamma_{i,j}$. For simplicity, we will use $\nabla L_i(\theta_j)$ to denote $\nabla_{\theta_j} L_i(\theta_j)$ in the remaining of the paper. The dual of Eq. (4) is a min-norm problem over variable $\lambda \in \Delta^n$ as follows:

$$\min_{\lambda \in \Delta^n} \left\| \sum_{i \in [n]} \lambda_i \Gamma_{i,j} \nabla L_i(\theta_j) \right\|^2, \quad \forall j \in [m], \tag{5}$$

which can be solved by existing Frank-Wolfe algorithms (Fujishige, 1980). Given the optimal dual solution $\lambda^*$ from Eq. (5), the primal solution of $d_j$ ($j \in [m]$) to Eq. (4) can be derived by the following convex combination of the $\Gamma$-weighted gradients:

$$d_j = \sum_{i \in [n]} \lambda_i^* \Gamma_{i,j} \nabla L_i(\theta_j). \tag{6}$$

To understand the benefits of Eq. (5), we consider classic MGDA: $\min_{\lambda \in \Delta^n} \left\| \sum_{i \in [n]} \lambda_i \nabla L_i(\theta) \right\|^2$, where $\lambda$ may be biased towards the objective with a small gradient (i.e., a well-optimized objective). In MosT, OT in Eq. (3) tends to result in a large $\Gamma_{i,j}$ for a small $L_i(\theta_j)$, thus moving small gradient away from the origin in Eq. (5) (i.e., preventing a well-optimized objective from dominating $d_j$).

The MGDA direction $d_j$ guarantees that every objective with non-zero $\Gamma_{i,j}$ will be improved or remain the same after updating $\theta_j$. After obtaining $d_j$, we update the model parameters $\theta_j$ by moving along this direction (Line 8 in Algorithm 1). Optionally, we can also run such gradient descent steps for $K$ steps in practice under the same $\Gamma$. In our convergence analysis (Section 4), we allow for $K > 1$ and assume a full batch setting with $\nabla L(\theta_j)$ evaluated on all the local data of problem $i$, for all $j$'s. Empirically, we report our experiment results based on mini-batch gradients in Section 5.

### 3.1 EXTENSION TO FEW-OBJECTIVE CASES

The MosT formulation discussed in the previous section is mainly motivated by the challenges of having many objectives in MOO. When $n \gg m$, the diversity of the $m$ models can be achieved by enforcing the two marginal constraints in the optimal transport problem. However, the diversity cannot be fully guaranteed when $n \ll m$. For example, when $n = 2$, by even applying uniform distributions for $\alpha$ and $\beta$ (i.e., the strongest constraints for diversity), a trivial but feasible solution of $\Gamma = [\mathbf{1}_{m/2}, \mathbf{0}_{m/2}; \mathbf{0}_{m/2}, \mathbf{1}_{m/2}]$ can collapse the $m$ models to duplicates of only two different models, i.e., one minimizing the first objective while the other minimizing the second. To address this problem, we create $(n' - n) \gg m$ dense interpolations of the $n$ objectives by sampling $(n' - n)$ groups of convex combination weights $w_{n+1:n'}$ on the simplex, i.e., $w_i \in \Delta^n$ drawn from a Dirichlet distribution. Then each auxiliary objective $L_i(\cdot)$ can be defined as

$$L_i(\theta) \triangleq \sum_{l \in [n]} w_{i,l} L_l(\theta), \quad \forall i = n+1, \cdots, n'. \tag{7}$$

Thereby, we increase the number of objectives to $n' \gg m$ and MosT can be applied to achieving diverse models for optimizing the $n'$ interpolations of the original $n$ objectives. This strategy can be explained as maximizing the coverage of the $m$ models over the dense samples of the Pareto front regions using $n'$ random reference vectors.

## 3.2 A Practical Solution-Specialization Curriculum

In scenarios with diverse objectives, each corresponding to a distinct domain or unique dataset, a practical demand arises: optimizing multiple models and turning them into a mixture of specialized experts (i.e., models). This allows each input sample select the best expert(s) for inference. Such "objective selecting expert/models" or "objective choice routing" strategy corresponds to removing $\Gamma\mathbf{1}_n = \beta$ from the constraint set $\Omega$ in Eq. (2). But it may lead to training imbalance among the $m$ models, e.g., one model is chosen by most objectives while other models get nearly zero optimization. As training proceeds, the winning model(s) trained by more objectives tend to be chosen even more frequently; hence joint optimization of $m$ models can collapse to training one single model.

To address this challenge, we propose to design a curriculum of varying the marginal constraints that progressively changes $\alpha$ and $\beta$ for different training stages. Specifically, in the early stage, we mainly focus on enforcing a uniform marginal distribution $\beta$ so that every model can receive sufficient training from multiple objectives. By relaxing the other marginal constraint $\alpha$ over $n$ objectives to be slightly non-uniform, the $m$ models have more degree of freedom to choose the objectives on which they perform the best (i.e., "model selecting objective" or "model choice routing") and we allow for slight imbalance among objectives.[3] During later stages, the curriculum instead focuses more on enforcing the marginal distribution $\alpha$ to be uniform so that every objective has to be covered by sufficient models. On the other hand, the marginal constraint $\beta$ can be relaxed in this stage since they are close to convergence. Empirical results in Appendix C.2 demonstrate the effectiveness of our curriculum strategy scheduling $\alpha$ and $\beta$.

## 4 Convergence Analyses

In this section, we analyze the convergence of MosT in Algorithm 1 for both strongly-convex and non-convex functions. The alternate minimization scheme poses additional challenges to our analysis compared with prior convergence results in MOO (Fliege et al., 2019). We show that our proposed algorithm using full gradients can converge to Pareto stationary points at a linear rate in the strongly-convex cases. We first make a common assumption.

**Assumption 1.** *Each objective $L_i(\theta)$ ($i \in [n]$) is $\nu$-smooth and $\mu$-strongly convex w.r.t. $\theta \in \mathbb{R}^d$.*

Our convergence result is as follows.

**Theorem 1** (Strongly-Convex)**.** *Let Assumption 1 holds. Given marginal distribution constraints $\alpha \in \Delta^n$ and $\beta \in \Delta^m$, under a fixed learning rate $\eta \leq \frac{1}{\nu}$, after running Algorithm 1 for $T$ outer iterations with full multi-gradient descent, we have for each solution $j \in [m]$,*

$$\left\|\theta_j^T - \theta_j^*\right\|^2 \leq (1 - \mu\eta\epsilon)^{TK} \left\|\theta_j^0 - \theta_j^*\right\|^2, \tag{8}$$

*where $\theta_j^*$ is a Pareto stationary solution and $\epsilon$ is a constant.*

A complete proof is provided in Appendix B. Our result is based on using full gradients of each objective when solving the min-norm problem (Eq. (4)). In practical implementation, we use $K > 1$ to run multiple iterations to update the model parameters for each objective locally, and use stochastic mini-batch gradients in Eq. (4). Note that $\theta_j^*$ here is some Pareto Stationary model of the objectives in Eq. (1). One limitation of our analyses is that we do not directly prove that the solutions achieved are more diverse than MGDA solutions.

**Theorem 2** (Non-Convex)**.** *Assume each objective $L_i(\theta)$ is $\nu$-smooth. Given marginal distribution constraints $\alpha \in \Delta^n$ and $\beta \in \Delta^m$, under a learning rate $\eta = \frac{1}{2\nu}$, after running Algorithm 1 for $T$ outer iterations with full batch multi-gradient descent, we have that*

$$\frac{1}{T} \sum_{t \in [T]} \sum_{j \in [m]} \beta_j \|d_j^t\|^2 \leq O\left(\frac{\nu}{T}\right). \tag{9}$$

Note that in the non-convex case, we prove the convergence rate of our proposed algorithm, which is the same as that of normal gradient descent under a fixed learning rate, but we do not show the Pareto optimality of our solutions.

---

[3]Less-selected objectives can be more difficult and it is more desirable to learn them later when models become more powerful.

## 5 MosT Applications

### 5.1 Experimental Setup

This section provides an overview of the experimental setup for our applications, covering the general baselines, evaluation metrics, and hyperparameters. The specific setup for each application is discussed in their respective sections, and more details can be found in Appendix D.

**Baselines.** For all applications, we compare with the following baselines.

- Linearization: Linearization-based MOO where we optimize over a convex combination of all objectives with $m$ randomly-sampled sets of weights. Minimizing a simple average of all the losses (empirical risk minimization over objectives) is a specific instance of this with uniform weights.
- MGDA: Running MDGA (Désidéri, 2012) independently for $m$ times with different random seeds.

In addition, we compare with task-specific algorithms that will be introduced with each application.

**Evaluation metrics.** For applications with a large number of objectives (large $n$), we use task-specific evaluation metrics, such as average accuracy (or tail accuracy) across all clients in FL. For applications with few objectivs (small $n$), we use the Hypervolume, a widely used metric for evaluating the quality of MOO solutions and is a proxy of diversity (Zitzler & Thiele, 1999). Hypervolume is feasible to compute when the number of objectives is small. Given a solution set $S \subset \mathbb{R}^n$ and a set of reference points $r = [r_1, \ldots, r_n] \subset \mathbb{R}^n$, the Hypervolume of $S$ measures the region weakly dominated by $S$ and bounded above by $r$: $H(S) = \Lambda \left( \{ q \in \mathbb{R}^n \mid \exists p \in S : p \leq q \text{ and } q \leq r \} \right)$, where $\Lambda(\cdot)$ denotes the Lebesgue measure. To ensure fair calculation of it, the reference points are kept consistent across all algorithms for each dataset. These reference points are determined either by following the settings of previous studies or by setting them as the upper bounds of the objective values from all algorithms to be compared. For experiments, we repeat each run for three times using different random seeds, and report the average and standard deviation of the multiple results.

**Hyperparameters.** For fair comparisons and clear visualization, we set the number of solutions $m = 5$. We tune the best learning rate via grid search for each method and each application. We set $\epsilon$ in Algorithm 3 as an adjustable hyperparameter. For the extended version of MosT (denoted as MosT-E) described in Section 3.1, we introduce additional hyperparameters $\alpha_1, \ldots, \alpha_n$, and $n'$ to handle the extension of existing objectives. The parameter $\alpha_i$ represents the positive shape parameter of the Dirichlet distribution, used to generate diverse objective weights, and $n'$ represents the number of extended objectives. We describe the detailed hyperparameter values in Appendix D.

### 5.2 Toy problems

We first demonstrate the effectiveness of MosT on a toy ZDT problem set. It is a popular MOO benchmark containing two objectives ($n = 2$) with oracle Pareto fronts (Zitzler et al., 2000). Specifically, we use ZDT-1, ZDT-2, and ZDT3, which are problems with 30 variables and exhibit convex, concave, and disconnected Pareto-optimal fronts, respectively. We compare MosT with two baselines described in Section 5.1 (denoted as 'linearization' and 'MGDA' below), and two additional methods—Exact Pareto Optimization (EPO) based on different preference vectors (Mahapatra & Rajan, 2020), and SVGD based on stein variational gradient descent (Liu et al., 2021b).

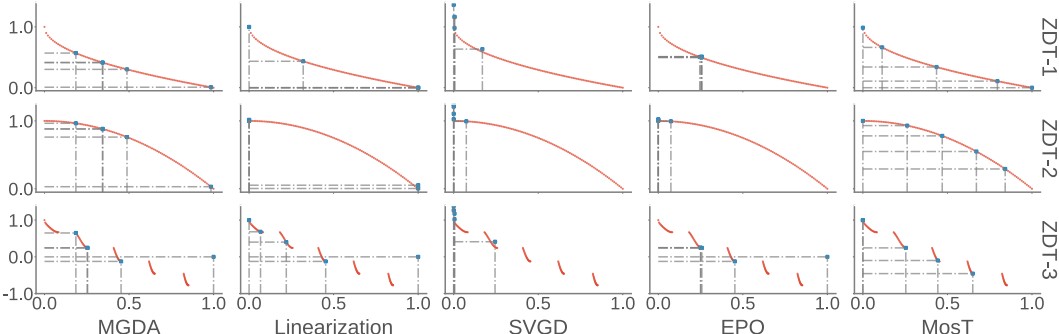

Figure 1: Solutions derived by different methods (blue scatters) on the ZDT bi-objective task, with the oracle Pareto-optimal fronts for the two objectives shown in red scatters.

Table 1: MosT achieves higher Hypervolumes than the baselines on the ZDT bi-objective problem.

| | MGDA | Linearization | SVGD | EPO | MosT |
|---|---|---|---|---|---|
| ZDT-1 | $4.02_{\pm 0.92}$ | $5.72_{\pm 0.01}$ | $5.54_{\pm 0.12}$ | $4.40_{\pm 0.01}$ | $\mathbf{5.84}_{\pm 0.02}$ |
| ZDT-2 | $4.63_{\pm 0.94}$ | $6.65_{\pm 0.00}$ | $6.65_{\pm 0.00}$ | $6.65_{\pm 0.00}$ | $\mathbf{6.87}_{\pm 0.02}$ |
| ZDT-3 | $4.53_{\pm 0.83}$ | $6.27_{\pm 0.02}$ | $5.77_{\pm 0.15}$ | $4.53_{\pm 0.68}$ | $\mathbf{6.36}_{\pm 0.04}$ |

We report the Hypervolumes of each method in Table 1 and visualize the obtained solutions alongside the entire Pareto-optimal fronts in Figure 1 for a more intuitive comparison. MosT leads to higher Hypervolumes, indicating its superior ability to generate more diverse solution sets that cover larger areas. Further analysis of the Pareto fronts reveals the following observations: 1) EPO and SVGD prioritize reducing one loss, potentially resulting in biased trade-offs, with SVGD lacking guaranteed convergence to Pareto-optimal solutions; 2) MGDA produces diverse solutions but fails to cover the entire Pareto-optimal fronts; 3) Linearization-based MOO is a competitive baseline with high Hypervolumes, but its solutions do not provide satisfactory diverse trade-offs, as evident from the Pareto fronts; 4) In contrast, MosT generates evenly-distributed solutions across the Pareto fronts.

## 5.3 FEDERATED LEARNING ($n \gg m$)

One important scenario where $n \gg m$ is the cross-device federated learning application, where we jointly learn $m$ models over a heterogeneous network of $n$ remote devices. The devices generate local data following non-identical distributions; hence we view the finite sum of empirical losses on each device as one objective, i.e., $L_i(\theta) := \frac{1}{v_i} \sum_{s=1}^{v_i} l_s(\theta)$ where $v_i$ is the number of local samples on device $i \in [n]$, and $l_s$ denotes the individual local loss on sample $s$. MosT seamlessly integrates with the decentralized setting of federated learning by computing client-specific local updates ($\theta_{1:m}$) and aggregating them to update the global model by $\Gamma$. Moreover, since clients are diverse, it is expected that the solution diversity benefits of MosT will contribute significantly to the final performance.

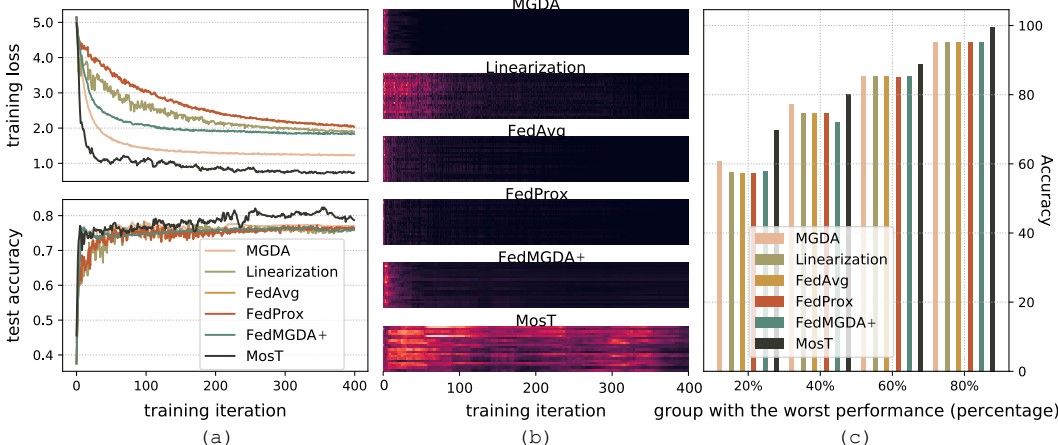

Figure 2: (a) Training loss and test accuracy curves of each method. MosT demonstrates faster convergence with higher accuracy. (b) Visualization of solution diversity during training, with each block on a column representing the KL distance of a pair of solutions (brighter indicates larger value). MosT produces more diverse solutions. (c) Accuracy of the worst 20%, 40%, 60% and 80% client groups. Diversity leads to better tail performance among all the objectives.

We conduct experiments on synthetic data and Federated Extended MNIST (FEMNIST) (Cohen et al., 2017; Caldas et al., 2018), where the number of objectives $n = 30$ and $n = 205$, respectively. We experiment on three synthetic datasets, denoted as Syn ($\rho_1, \rho_2$), with different $\rho_1$ and $\rho_2$ controlling heterogeneity of local models and data, as detailed in Appendix D. We compare MosT with baselines described in Section 5.1 and state-of-the-art federated learning algorithms, including FedAvg (McMahan et al., 2017), FedProx (Li et al., 2020b), and FedMGDA+ (Hu et al., 2022). We run the threeeach algorithm $m$ times with different random initializations. It is worth noting that during evaluation, for all methods, we let each device pick a best model out of $m$ solutions based on the validation set, and compute its performance. We then report the average and the quantile accuracy across all devices. As the results shown in Table 2, MosT outperforms the baselines by a large margin on all datasets. Furthermore, we have the following observations.

Table 2: Average accuracy across all clients (mean and std across 3 runs) on synthetic datasets and FEMNIST. MosT outperforms the strong baselines.

|  | MGDA | Linearization | FedAvg | FedProx | FedMGDA+ | MosT |
|---|---|---|---|---|---|---|
| Syn (0.0, 0.0) | $77.22_{\pm 0.41}$ | $75.91_{\pm 0.37}$ | $75.71_{\pm 0.51}$ | $75.60_{\pm 0.42}$ | $75.26_{\pm 1.21}$ | $\mathbf{83.09}_{\pm 0.87}$ |
| Syn (0.5, 0.5) | $87.09_{\pm 0.29}$ | $87.18_{\pm 0.27}$ | $86.26_{\pm 0.61}$ | $86.13_{\pm 0.39}$ | $85.21_{\pm 1.42}$ | $\mathbf{89.07}_{\pm 0.63}$ |
| Syn (1.0, 1.0) | $90.52_{\pm 0.13}$ | $89.87_{\pm 0.51}$ | $88.12_{\pm 0.75}$ | $87.58_{\pm 1.36}$ | $87.16_{\pm 1.09}$ | $\mathbf{91.70}_{\pm 0.02}$ |
| FEMNIST | $78.86_{\pm 1.43}$ | $68.80_{\pm 0.87}$ | $68.75_{\pm 0.47}$ | $68.92_{\pm 0.41}$ | $80.08_{\pm 0.12}$ | $\mathbf{80.94}_{\pm 0.34}$ |

**MosT results in significant convergence improvements.** Figure 2(a) compares the training loss and test accuracy curves of different algorithms. Notably, MosT outperforms the baselines with a lower training loss (faster convergence) and higher test accuracy (better generalization to unseen test data).

**MosT maintains diversity throughout the training process.** To evaluate this, we analyze how the diversity of solutions evolves for different algorithms. We quantify diversity by calculating the Kullback Leibler (KL) distance between predictions of any pair of solutions generated by the same algorithm. Figure 2(b) shows that initially, all algorithms exhibit high diversity due to different weight initialization. However, within the baselines, solution diversity decreases significantly during training. In contrast, MosT maintains high diversity throughout the training process.

**MosT promotes fairness in FL.** The significant improvement in diversity achieved by MosT is expected to benefit clients who are often overlooked by other algorithms, leading to greater fairness. To validate this, we investigate the accuracy of the worst 20%, 40%, 60%, and 80% clients, and compare across all algorithms. As depicted in Figure 2(c), MosT outperforms the baselines by an larger margin for clients with worse performance, which demonstrates that the diversity of MosT effectively promotes fairness in FL.

Furthermore, our study reveals that MosT assigns clients with diverse solutions for inference. A detailed analysis of this finding can be found in Section 5.5.

## 5.4 FAIRNESS-ACCURACY TRADE-OFFS ($n \ll m$)

In this section, we apply MosT to explore various trade-offs between accuracy and algorithmic fairness (i.e., statistical independence between predictions and sensitive attributes). The number of objectives $n$ is 2. However, using optimal transport to match solutions and objectives in this scenario may produce feasible but trivial solutions as explained in Section 3.1. Hence, we adapt the extension of MosT named MosT-E (introduced in Section 3.1).

As discussed in Section 2, prior works that address fairness-accuracy trade-offs can be limited due to the difficulty of setting constraints before training (Zafar et al., 2017), or the mismatch between diverse exploration space and diverse solutions (Mahapatra & Rajan, 2020). MosT-E differs by sampling a wide range of preference vectors to encompass various trade-offs comprehensively and using optimal transport to automatically generate solutions that maximize coverage for all preference vectors. We quantify the fairness objective using *disparate impact* (Court, 1971), and optimize it using its convex approximation (Zafar et al., 2017). We experiment on a synthetic dataset (Zafar et al., 2017) and a real German credit dataset (Asuncion & Newman, 2007). We compare MosT and MosT-E with MGDA, linearization-based MOO (which can be viewed as a soft version of Zafar et al. (2017), and EPO (Mahapatra & Rajan, 2020), and select the best parameters for each method based on the highest Hypervolume on a validation set.

Table 3: Hypervolumes ($\times 100$) on 5 solutions with different fairness-accuracy trade-offs. MosT-E achieves the highest Hypervolume coverage on two (fairness, accuracy) objectives.

|  | MGDA | Linearization | EPO | MosT | MosT-E |
|---|---|---|---|---|---|
| Synthetic | $4.21_{\pm 1.10}$ | $6.57_{\pm 0.02}$ | $2.43_{\pm 0.01}$ | $4.85_{\pm 0.96}$ | $\mathbf{7.24}_{\pm 0.33}$ |
| German | $1.95_{\pm 0.01}$ | $1.92_{\pm 0.02}$ | $1.87_{\pm 0.03}$ | $1.97_{\pm 0.32}$ | $\mathbf{2.05}_{\pm 0.00}$ |

**MosT-E generates more diverse trade-offs.** Table 3 shows that MosT-E achieves the highest Hypervolumes, suggesting a superior quality of the solution set it generates. Furthermore, Figure 3 demonstrates that MosT-E generates solutions that are not only more diverse but also more evenly distributed across the Pareto fronts. We provide results for increased solution size in Appendix C.1, which further supports MosT-E is capable of effectively accommodating various preferences.

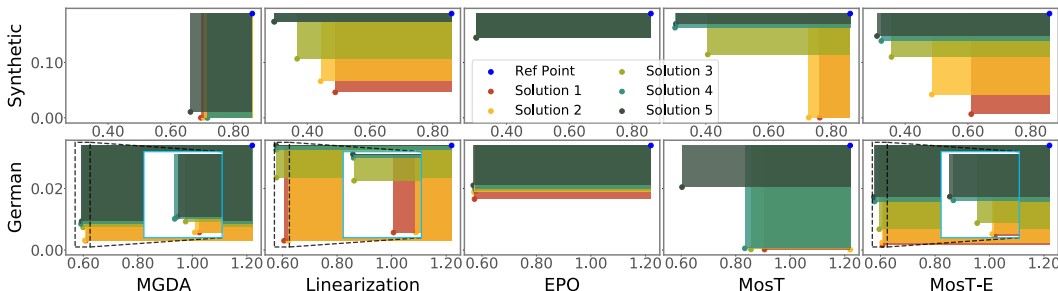

Figure 3: Hypervolumes (colored areas) formed by five solutions for classification loss (objective 1, x-axis) and fairness (objective 2, y-axis) on synthetic and German datasets. For the German dataset, we zoom in on specific areas. We see that MosT and MosT-E cover larger areas than the baselines.

**MosT-E effectively addresses the problem of MosT under $n \ll m$.** When $n \ll m$, MosT may assign models separately to dominate individual objectives, resulting in solutions without sufficient diversity. The solutions generated by MosT shown in Figure 3, align with our idea by predominantly prioritizing either low classification loss or low disparate impact. This limitation is effectively overcome by MosT-E, with diversely combining existing few objectives as new objectives.

## 5.5 ABLATION STUDY: PREVENTING COLLAPSE WITH OPTIMAL TRANSPORT MECHANISM

In this section, we conduct ablation studies on the effectiveness of the optimal transport matching (Eq. (2)). We compare three strategies introduced in Section 3.2: 1) the original MosT objective, which utilizes optimal transport; 2) "objective selecting model", which selects the best expert/model for each objective (i.e., removing the $\Gamma^\top \mathbf{1}_n = \beta$ constraint); and 3) "model selecting objective", which selects the best objective for each model (i.e., removing the $\Gamma \mathbf{1}_m = \alpha$ constraint).

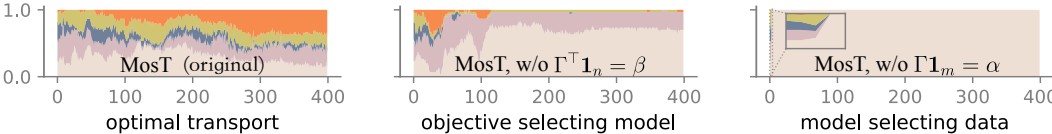

Figure 4: The percentage of assigned objectives for each model under three matching strategies. Each color band represents a model, with the y-axis indicating the corresponding percentage. We see that MosT (left) learns 5 diverse models that servers the 30 objectives in a balanced manner.

To ensure a fair comparison, we initialize with the same model weights. Throughout the training process, we track the assignment of objectives to each model, i.e., for every model, identifying the objectives with the smallest validation loss. We visualize the percentages of the selected objectives for each model over time in Figure 4 on the Syn (0.0, 0.0) dataset. In the case of "objective selecting model" (middle), we observe that two of the models progressively dominate all the objectives. Similarly, "model selecting objective" shows the early dominance of one model. These observations confirm the presence of the collapse phenomenon in MOO, where limited solutions dominate all objectives. On the contrary, MosT using optimal transport involving a two-way matching shows a more balanced distribution of objectives among the models throughout training.

## 6 CONCLUSIONS

In this paper, we have proposed "multi-objective multi-solution transport (MosT)", a framework that aims to find $m$ Pareto solutions (models) that achieve diverse trade-offs among $n$ optimization objectives. We have specifically investigated a challenging case of $n \gg m$, in which existing methods often struggle with exploring a high-dimensional Pareto frontier. We formulate MosT as a bi-level optimization of multiple weighted objectives, where the weights guide the exploration and are determined by an optimal transport (OT) matching objectives and solutions. Our algorithm theoretically converges to $m$ Pareto solutions by alternating between optimizing the weighted objectives and OT. MosT can be extended to achieve diverse solutions for the $n \ll m$ cases. We have applied MosT to a rich class of machine learning problems that involve training $m$ models to serve $n$ users, domains, or criteria. Empirically, we have observed that MosT outperforms other strong baselines in tasks including federated learning, fairness-accuracy trade-offs, and other simpler MOO benchmarks.

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

## A  BACKGROUND ON MGDA IN MULTI-OBJECTIVE OPTIMIZATION

We first describe some background on the multi-gradient descent algorithm to solve multi-objective optimization.

Let $\mathbf{L}(\theta) \in \mathbb{R}^n$ be defined as

$$\mathbf{L}(\theta) := (L_1(\theta), \cdots, L_n(\theta)), \theta \in \mathbb{R}^d. \tag{10}$$

The goal for the multi-objective optimization (minimization) problem is to find Pareto optimal solutions with respect to all objectives $L_i(\cdot), i \in [n]$. One line of method is at each iteration, to find a common descent direction $d$ for all objectives. Given the current model $\theta$, we would like to find a descent step to minimize each objective value. For the single-objective case, the direction is $-\nabla L(\theta)$. For $n$ objectives, one objective is to solve for $d$:

$$\min_d \left\{ \max_{i \in [n]} \nabla L_i(\theta)^\top d + \frac{1}{2} \|d\|^2 \right\}, \tag{11}$$

and then apply $d$ as $\theta = \theta + \eta d$. If the optimal objective value of Eq. (11) is negative, then there exists a descent direction $d^*$ such that all objective values will be decreased. If $\theta$ is Pareto stationary, then $d = \mathbf{0}$ and the optimal objective value is 0. This formulation is equivalent to

$$\min_{b,d} \quad b + \frac{1}{2} \|d\|^2 \tag{12}$$

$$s.t. \quad \nabla L_i(\theta)^\top d \le b, i \in [n].$$

Formally, we have the following lemma.

**Lemma 1** (Good Descent Direction Désidéri (2012) ). *Let $d, b$ be the solutions of Eq. (12), then*

1. *If $\theta$ is Pareto stationary, then $d = \mathbf{0}$ and $b = 0$.*

2. *If $\theta$ is not Pareto stationary, then*

$$b \le -\frac{1}{2}\|d\|^2 < 0, \tag{13}$$

$$\nabla L_i(\theta)^\top d \le b, \ i \in [n]. \tag{14}$$

**Lemma 2** (A Rescaled Version of Lemma 1). *Let $d_j \in \mathbb{R}^d$ be the solution of Eq. (4) and $\Gamma_{i,j}$ be some non-negative scalar, then*

1. *If $\theta_j$ is Pareto stationary, then $d_j = \mathbf{0}$.*

2. *If $\theta_j$ is not Pareto stationary, then*

$$\Gamma_{i,j} L_i(\theta_j)^\top d_j \le -\frac{1}{2}\|d_j\|^2, \ i \in [n]. \tag{15}$$

## B  CONVERGENCE PROOFS

### B.1  STRONGLY-CONVEX CASES

First, let us assume $K = 1$. At each iteration $t$, we have

$$d_j^t = -\sum_{i \in [n]} \lambda_i \Gamma_{i,j}^t \nabla L_i(\theta_j^t), \ \forall j \in [m] \tag{16}$$

for some $\{\lambda_i\}_{i \in [n]} \in \Delta^n$ which is the solution of Eq. (5). First we note that for every $j \in [m]$, $\theta_j$ will converge. If $d_j^t = \mathbf{0}$, then $\theta_j$ has converged to a Pareto stationary point. Otherwise, for every objective $L_i$, we have sufficient decrease: $L_i(\theta_j^{t+1}) - L_i(\theta_j^t) \le -\frac{1}{2}\eta(1 - \nu\eta)\|d_j^t\|^2 < 0$. Therefore, every solution will converge. We denote $\theta_j^*$ as one of the Pareto stationary solutions of Eq. (1) that solution $\theta_j$ converges to. By definition and the properties of MDGA, we know that for every solution,

every objective value will be non-increasing throughout optimization. Hence, for every $i \in [n]$ and $j \in [m]$, it holds that

$$L_i(\theta_j^{t+1}) - L_i(\theta_j^*) \geq 0. \tag{17}$$

For every $i \in [n]$, the $\nu$-smoothness and $\mu$-convexity of $L_i$ lead to

$$L_i(\theta_j^{t+1}) = L_i(\theta_j^t + \eta d_j^t) \tag{18}$$

$$\leq L_i(\theta_j^t) + \eta \nabla L_i(\theta_j^t)^\top d_j^t + \frac{\nu}{2} \|\eta d_j^t\|^2 \tag{19}$$

$$\leq L_i(\theta_j^*) + \nabla L_i(\theta_j^t)^\top (\theta_j^t - \theta_j^*) - \frac{\mu}{2} \|\theta_j^t - \theta_j^*\|^2 + \eta \nabla L_i(\theta_j^t)^\top d_j^t + \frac{\nu}{2} \|\eta d_j^t\|^2. \tag{20}$$

By moving $L_i(\theta_j^*)$ to the left-hand side and multiplying both sides by $\lambda_i \Gamma_{i,j}^t$, we have

$$\sum_{i \in [n]} \lambda_i \Gamma_{i,j}^t \left( L_i(\theta_j^{t+1}) - L_i(\theta_j^*) \right) \tag{21}$$

$$\leq \sum_{i \in [n]} \lambda_i \Gamma_{i,j}^t \nabla L_i(\theta_j^t)^\top (\theta_j^t - \theta_j^* + \eta d_j^t) - \sum_{i \in [n]} \lambda_i \Gamma_{i,j}^t \frac{\mu}{2} \|\theta_j^t - \theta_j^*\|^2 + \sum_{i \in [n]} \lambda_i \Gamma_{i,j}^t \frac{\nu}{2} \|\eta d_j^t\|^2. \tag{22}$$

As $\Gamma_{i,j}^t \geq \epsilon > 0$, then

$$\sum_{i \in [n]} \lambda_i \Gamma_{i,j}^t \frac{\mu}{2} \|\theta_j^t - \theta_j^*\|^2 \leq \frac{\mu \epsilon}{2} \|\theta_j^t - \theta_j^*\|^2. \tag{23}$$

Due to the Hölder inequality, we have $\sum_{i \in [n]} \lambda_i \Gamma_{i,j} \leq \|\lambda\|_1 \|\Gamma_{\cdot,j}\|_\infty := \beta_j \leq 1$. Hence, we have

$$\sum_{i \in [n]} \lambda_i \Gamma_{i,j}^t \left( L_i(\theta_j^{t+1}) - L_i(\theta_j^*) \right)$$

$$\leq \sum_{i \in [n]} \lambda_i \Gamma_{i,j}^t \nabla L_i(\theta_j^t)^\top (\theta_j^t - \theta_j^* + \eta d_j^t) - \frac{\mu \epsilon}{2} \|\theta_j^t - \theta_j^*\|^2 + \frac{\nu \beta_j}{2} \|\eta d_j^t\|^2 \tag{24}$$

$$= -d_j^t(\theta_j^t - \theta_j^*) - \eta \|d_j^t\|^2 - \frac{\mu \epsilon}{2} \|\theta_j^t - \theta_j^*\|^2 + \frac{\nu \beta_j}{2} \|\eta d_j^t\|^2$$

$$\leq -d^t(\theta_j^t - \theta_j^*) - \eta \left( 1 - \frac{\beta_j}{2} \right) \|d_j^t\|^2 - \frac{\mu \epsilon}{2} \|\theta_j^t - \theta_j^*\|^2 \quad \text{(taking } \eta \leq \frac{1}{\nu}\text{)} \tag{25}$$

$$\leq -d^t(\theta_j^t - \theta_j^*) - \frac{\eta}{2} \|d_j^t\|^2 - \frac{\mu \epsilon}{2} \|\theta_j^t - \theta_j^*\|^2 \quad \text{(using } \beta_j \leq 1\text{)}$$

$$= -\frac{1}{2\eta} (2\eta d_j^t(\theta_j^t - \theta_j^*) + \|\eta d_j^t\|^2) - \frac{\mu \epsilon}{2} \|\theta_j^t - \theta_j^*\|^2$$

$$= -\frac{1}{2\eta} (2(\theta_j^{t+1} - \theta_j^t)^\top (\theta_j^t - \theta_j^*) + \|\theta_j^{t+1} - \theta_j^t\|^2) - \frac{\mu \epsilon}{2} \|\theta_j^t - \theta_j^*\|^2$$

$$= -\frac{1}{2\eta} (2\theta_j^{t+1}\theta_j^t - 2\|\theta_j^t\|^2 - 2(\theta_j^{t+1} - \theta_j^t)^\top \theta_j^* + \|\theta_j^{t+1}\|^2 + \|\theta_j^t\|^2 - 2\theta_j^{t+1}\theta_j^t) - \frac{\mu \epsilon}{2} \|\theta_j^t - \theta_j^*\|^2$$

$$= -\frac{1}{2\eta} (\|\theta_j^{t+1}\|^2 - 2(\theta_j^{t+1} - \theta_j^t)^\top \theta_j^* - \|\theta_j^t\|^2) - \frac{\mu \epsilon}{2} \|\theta_j^t - \theta_j^*\|^2$$

$$= -\frac{1}{2\eta} (\|\theta_j^{t+1} - \theta_j^*\|^2 - \|\theta_j^t - \theta_j^*\|^2) - \frac{\mu \epsilon}{2} \|\theta_j^t - \theta_j^*\|^2$$

$$= \frac{1}{2\eta} (\|\theta_j^t - \theta_j^*\|^2 - \|\theta_j^{t+1} - \theta_j^*\|^2) - \frac{\mu \epsilon}{2} \|\theta_j^t - \theta_j^*\|^2. \tag{26}$$

Since during optimization, we guarantee that every objective value will be non-increasing at each iteration, we have $L_i(\theta_j^{t+1}) - L_i(\theta_j^*) \geq 0$. So the left-hand side of Eq. (21) is non-negative. Hence,

$$\frac{1}{2\eta} (\|\theta_j^t - \theta_j^*\|^2 - \|\theta_j^{t+1} - \theta_j^*\|^2) - \frac{\mu \epsilon}{2} \|\theta_j^t - \theta_j^*\|^2 \geq 0, \tag{27}$$

$$\|\theta_j^{t+1} - \theta_j^*\|^2 \leq (1 - \mu \eta \epsilon) \|\theta_j^t - \theta_j^*\|^2, \tag{28}$$

which gives us linear convergence.

When $K > 1$, at each outer iteration, fixing $\Gamma_{i,j}^t$, we are running multiple updates on the model parameters. In this case, we still have

$$\|\theta_j^{t,k+1} - \theta_j^*\| \leq (1 - \mu\eta\epsilon)\|\theta_j^{t,k} - \theta_j^*\|^2, \tag{29}$$

where $\|\theta_j^{t,k+1}\|$ denote the model parameters at the $t$-th outer iteration and $k + 1$-th inner iteration (Line 6 of Algorithm 1). Hence,

$$\|\theta_j^{t+1} - \theta_j^*\| \leq (1 - \mu\eta\epsilon)^K \|\theta_j^t - \theta_j^*\|^2 \tag{30}$$

holds.

## B.2 Non-Convex and smooth cases

For simplicity, we first consider the case where $K = 1$. From Lemma 2, we know that at each iteration $t$,

$$\Gamma_{i,j}^t \nabla L_i(\theta_j^t)^\top d_j^t \leq -\frac{1}{2} \left\| d_j^t \right\|^2, \quad \forall i \in [n]. \tag{31}$$

Assuming $\nu$-smooth of each $L_i$, we have

$$\Gamma_{i,j}^t \left( L_i(\theta_j^{t+1}) - L_i(\theta_j^t) \right) = \Gamma_{i,j}^t \left( L_i(\theta_j^t + \eta d_j^t) - L_i(\theta_j^t) \right) \tag{32}$$

$$\leq \eta \Gamma_{i,j}^t \nabla L_i(\theta_j^t)^\top d_j^t + \frac{\nu \Gamma_{i,j}^t}{2} \|\eta d_j^t\|^2 \tag{33}$$

$$\leq -\frac{\eta}{2}\|d_j^t\|^2 + \frac{\nu\eta^2}{2} \Gamma_{i,j}^t \|d_j^t\|^2 \tag{34}$$

$$\leq -\frac{\eta}{2} \Gamma_{i,j}^t \|d_j^t\|^2 + \frac{\nu\eta^2}{2} \Gamma_{i,j}^t \|d_j^t\|^2 \quad (\Gamma_{i,j}^t \leq 1) \tag{35}$$

$$= -\frac{\eta(1 - \nu\eta)}{2} \Gamma_{i,j}^t \|d_j^t\|^2. \tag{36}$$

Sum over all models $j \in [m]$,

$$\sum_{j \in [m]} \Gamma_{i,j}^t \left( L_i(\theta_j^{t+1}) - L_i(\theta_j^t) \right) \leq -\frac{\eta(1 - \nu\eta)}{2} \sum_{j \in [m]} \Gamma_{i,j}^t \|d_j^t\|^2. \tag{37}$$

The above result is for a single objective $L_i(\cdot)$. Now let's consider the weighted sum of all the $n$ objectives between two steps with different $\Gamma$'s, i.e., $\Gamma^t$ and $\Gamma^{t+1}$. We have

$$\sum_{i \in [n]} \sum_{j \in [m]} \left( \Gamma_{i,j}^{t+1} L_i(\theta_j^{t+1}) - \Gamma_{i,j}^t L_i(\theta_j^t) \right) \tag{38}$$

$$\leq \sum_{i \in [n]} \sum_{j \in [m]} \Gamma_{i,j}^t \left( L_i(\theta_j^{t+1}) - L_i(\theta_j^t) \right) \quad \text{(optimality of } \Gamma^{t+1} - \epsilon I) \tag{39}$$

$$\leq -\frac{\eta(1 - \nu\eta)}{2} \sum_{j \in [m]} \sum_{i \in [n]} \Gamma_{i,j}^t \|d_j^t\|^2 \quad \text{(apply Eq. (37))} \tag{40}$$

$$= -\frac{\eta(1 - \nu\eta)}{2} \sum_{j \in [m]} \beta_j \|d_j^t\|^2. \tag{41}$$

Here $\beta_j$ is the $j$-th dimension of $\beta \in \Delta^m$ in formulation Eq. (2). Hence,

$$\sum_{j \in [m]} \beta_j \|d_j^t\|^2 \leq \frac{2}{\eta(1 - \nu\eta)} \sum_{i \in [n]} \sum_{j \in [m]} \left( \Gamma_{i,j}^t L_i(\theta_j^t) - \Gamma_{i,j}^{t+1} L_i(\theta_j^{t+1}) \right). \tag{42}$$

Applying telescope sum on both sides gives

$$\sum_{t \in [T]} \sum_{j \in [m]} \beta_j \|d_j^t\|^2 \leq \frac{2}{\eta(1 - \nu\eta)} \sum_{i \in [n]} \sum_{j \in [m]} \left( \Gamma_{i,j}^1 L_i(\theta_j^1) - \Gamma_{i,j}^{T+1} L_i(\theta_j^{T+1}) \right) := C. \tag{43}$$

Then we get the following bound on the average gradient norm:

$$\frac{1}{T} \sum_{t \in [T]} \sum_{j \in [m]} \beta_j \|d_j^t\|^2 \leq \frac{C}{T}. \tag{44}$$

If we take $\nu\eta = \frac{1}{2}$, then $C = O(\nu)$. This gives us a $O\left(\frac{\nu}{T}\right)$ rate in terms of gradient norms for non-convex cases under a fixed learning rate.

For the case where $K > 1$, we have

$$\sum_{i \in [n]} \sum_{j \in [m]} \left(\Gamma_{i,j}^{t+1} L_i(\theta_j^{t+1}) - \Gamma_{i,j}^t L_i(\theta_j^t)\right) \leq -\frac{\eta(1-\nu\eta)}{2} \sum_{j \in [m]} \beta_j \sum_{k \in [K]} \left\|d_j^{t,k}\right\|^2 \tag{45}$$

$$\leq -\frac{\eta(1-\nu\eta)}{2} \sum_{j \in [m]} \beta_j \left\|d_j^t\right\|^2, \tag{46}$$

where $k$ denotes the index for inner updates on model parameters fixing $\Gamma^t$. Similarly, we have the result

$$\frac{1}{T} \sum_{t \in [T]} \sum_{j \in [m]} \beta_j \left\|d_j^t\right\|^2 \leq \frac{C}{T}. \tag{47}$$

# C    ADDITIONAL ANALYSIS

## C.1    SENSITIVITY ANALYSIS OF THE NUMBER OF SOLUTIONS ($m$)

In this section, we explore the impact of the solution set size $m$, on the quality of the generated solution set. To maintain consistent comparisons throughout the experiments in Section 5, we hold $m$ at 5. Increasing the number of generated solutions is expected to lead to a denser distribution of solutions across the Pareto-optimal fronts. To explore this, we conducted an ablation experiment on the synthetic fairness-accuracy trade-off dataset, varying $m$ from 5 to 10.

Table 4 summarizes Hypervolumes, illustrating that augmenting $m$ enhances solution set quality across all algorithms, with the most significant improvement observed in MosT. This indicates that increasing the number of solutions aids MosT in averting convergence to trivial solutions when $n \ll m$. However, this comes at a higher computational cost due to the need for extra model training. In contrast, MosT-E effectively addresses this issue without incurring excessive computational expenses. As depicted in Figure 5, when compared to MosT-E with $m = 5$, using $m = 10$ results in denser and more evenly distributed trade-offs.

Table 4: Experimental results of Hypervolumes ($\times 100$) on the synthetic fairness-accuracy trade-off dataset for $m = 5$ and $m = 10$.

|  | MGDA | Linearization | EPO | MosT | MosT-E |
|---|---|---|---|---|---|
| $m = 5$ | $4.21_{\pm 1.10}$ | $6.57_{\pm 0.02}$ | $2.43_{\pm 0.01}$ | $4.85_{\pm 0.96}$ | $\mathbf{7.24}_{\pm 0.33}$ |
| $m = 10$ | $5.01_{\pm 0.59}$ | $7.56_{\pm 0.01}$ | $4.00_{\pm 0.11}$ | $6.44_{\pm 0.19}$ | $\mathbf{8.00}_{\pm 0.05}$ |

## C.2    ABLATION STUDY FOR CURRICULUM LEARNING

Table 5: Experimental results of synthetic datasets of federated learning tasks comparing MosT w/ and w/o curriculum learning.

|  | MGDA | Linearization | FedAvg | FedProx | FedMGDA+ | MosT w/o CL | MosT |
|---|---|---|---|---|---|---|---|
| Syn (0.0, 0.0) | $77.22_{\pm 0.41}$ | $75.91_{\pm 0.37}$ | $75.71_{\pm 0.51}$ | $75.60_{\pm 0.42}$ | $75.26_{\pm 1.21}$ | $79.78_{\pm 0.25}$ | $\mathbf{83.09}_{\pm 0.87}$ |
| Syn (0.5, 0.5) | $87.09_{\pm 0.29}$ | $87.18_{\pm 0.27}$ | $86.26_{\pm 0.61}$ | $86.13_{\pm 0.39}$ | $85.21_{\pm 1.42}$ | $88.18_{\pm 0.41}$ | $\mathbf{89.07}_{\pm 0.63}$ |
| Syn (1.0, 1.0) | $90.52_{\pm 0.13}$ | $89.87_{\pm 0.51}$ | $88.12_{\pm 0.75}$ | $87.58_{\pm 1.36}$ | $87.16_{\pm 1.09}$ | $91.33_{\pm 0.27}$ | $\mathbf{91.70}_{\pm 0.02}$ |

In this ablation study, we evaluate the impact of curriculum learning (from Section 3.2) on optimizing multiple models using MosT.

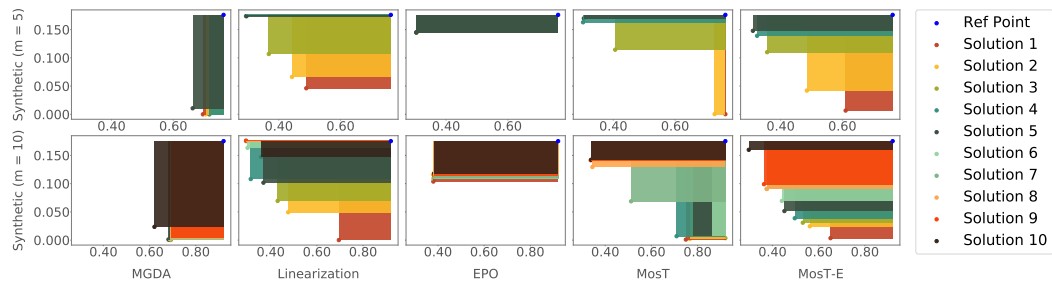

Figure 5: Pareto fronts obtained with $m = 5$ and $m = 10$ for classification loss (x-axis) and disparate impact (y-axis) on the synthetic dataset of fairness-accuracy trade-off task.

**Curriculum setup for MosT.** As introduced in Section 3.2, our proposed curriculum strategy involves adjusting marginal distributions $\alpha$ and $\beta$ over different training stages to balance the freedom of 'model selecting objective' and 'objective selecting model'. In the initial stages, we prioritize a uniform distribution for $\beta$ to ensure exposure to multiple objectives. As training progresses, we transition $\alpha$ to a uniform distribution, covering all objectives, while relaxing $\beta$. This transition is achieved by a hyperparameter that gradually decreases from 1 to 0. Though this hyperparameter gradually approaches zero, the transition direction differs: it shifts $\beta$ from uniform to performance-oriented and, conversely, shifts $\alpha$ in the opposite direction.

We compare standard MosT with a variant using uniform marginal distributions for $\alpha$ and $\beta$ throughout training, denoted as 'MosT w/o CL'. We hypothesize that curriculum learning enhances overall performance and training stability. We conduct experiments on three synthetic federated learning datasets, presenting results in Table 5. Curriculum learning significantly improves the performance of MosT, proving its effectiveness. Notably, even without curriculum learning, MosT outperforms other algorithms.

Furthermore, Figure 6 illustrates the training loss and test accuracy curves, highlighting the stability difference between the two approaches during training. Curriculum learning leads to increased stability and better convergence towards better solutions.

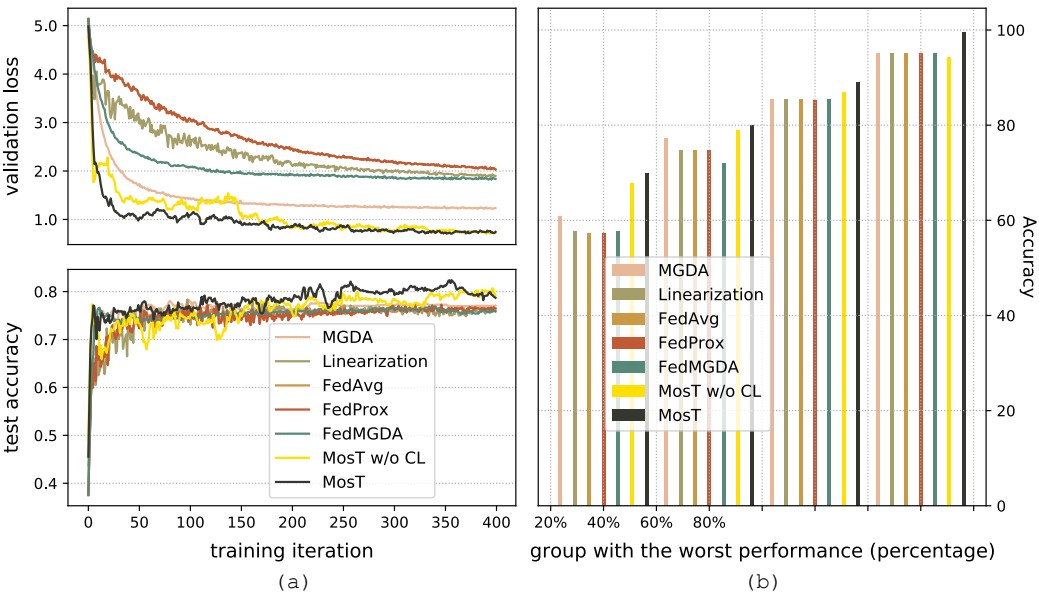

Figure 6: Including MosT w/o curriculum learning, (a) displays training loss and test accuracy curves; (b) shows the accuracy of the worst 20%, 40%, 60% and 80% client groups.

Table 6: Comparisons among MosT and its two linearization variants: MosT (L) and w-MGDA. The linearization methods involve taking a weighted sum of the model parameters trained on each objective for updates. In the case of w-MGDA, diverse weight vectors are employed to compute MGDA $m$ times.

| | MGDA | w-MGDA | Linearization | FedAvg | FedProx | FedMGDA+ | MosT (L) | MosT |
|---|---|---|---|---|---|---|---|---|
| Syn (0.0, 0.0) | $77.22_{\pm 0.41}$ | $77.96_{\pm 0.86}$ | $75.91_{\pm 0.37}$ | $75.71_{\pm 0.51}$ | $75.60_{\pm 0.42}$ | $75.26_{\pm 1.21}$ | $79.89_{\pm 0.93}$ | $\mathbf{83.09}_{\pm 0.87}$ |
| Syn (0.5, 0.5) | $87.09_{\pm 0.29}$ | $86.35_{\pm 0.38}$ | $87.18_{\pm 0.27}$ | $86.26_{\pm 0.61}$ | $86.13_{\pm 0.39}$ | $85.21_{\pm 1.42}$ | $86.98_{\pm 0.36}$ | $\mathbf{89.07}_{\pm 0.63}$ |
| Syn (1.0, 1.0) | $90.52_{\pm 0.13}$ | $89.37_{\pm 0.72}$ | $89.87_{\pm 0.51}$ | $88.12_{\pm 0.75}$ | $87.58_{\pm 1.36}$ | $87.16_{\pm 1.09}$ | $90.12_{\pm 1.13}$ | $\mathbf{91.70}_{\pm 0.02}$ |

### C.3 ABLATION STUDY ON MOST DESIGN

MosT relies on assignment generated by OT to find a balance matching between objectives and solutions, and then weight the original objectives to optimize domain experts using MGDA. In this section, we conduct ablation studies to verify the design of OT and weighted MGDA, respectively. Experiments are carried out on three synthetic federated learning datasets and detailed in Table 6.

**OT-Generated v.s. Randomly Generated Weight Assignments.** We compare the weight assignments generated by OT with the randomly generated weight assignments to verify the impact of the choice of objective weighting method in MGDA on the overall performance of MosT. In other word, we compare MosT with executing MGDA $m$ times with randomly generated weights, which is denoted as w-MGDA. Experimental results reveal OT-generated weights work significantly better than random weights. *This illustrates the necessity of using OT to find a balanced match between solutions and objectives.*

**MGDA v.s. Linearization in Weighted Multi-Objective Optimization.** We compare MosT that uses MGDA against the variant that updates model parameters solely based on the optimal transport solution weights. We aim to understand how effectively these two methods determine gradient updates for weighted multi-objective optimization. In the variant of MosT, denoted as MosT (L), instead of seeking the Pareto solution of $m$ weighted objectives (as indicated in Eq. (1)), we compute $\theta_j$ as $\theta_j = \sum_{i=1}^{n} \Gamma_j^i \theta_j^i$, where $\theta_j^i$ represents the parameter of $\theta_j$ trained on data from the $i$-th objective. The experimental results showcase the consistent superiority of MosT over both MGDA and MosT (L) across three synthetic federated learning scenarios. *It proves the effectiveness of employing MGDA for parameter updates.*

## D EXPERIMENTAL DETAILS

We will provide a detailed description of the models and hyperparameters employed for each dataset. It is important to note that all algorithms adhere to a common setup, which includes the train-validation-test split, the number of training epochs, and the tunable learning rates.

**Hyperparameters for MosT.** In Algorithm 3, we introduce a tunable hyperparameter denoted as $\epsilon$. It is defined as $\epsilon = \frac{k}{n}$, where $k$ is a selected from $\{2.0, 1.5, 1.0, 0.5\}$. Throughout the training process, we progressively decrease the value of $k$ to mitigate any adverse effects on solution diversity while maintaining $\epsilon$ above a predefined threshold.

**Hyperparameters for baselines.** In addition to the general setup, we fine-tune hyperparameters tailored to each baseline model, aligning with their original configurations. As an example, we adjust the hyperparameter responsible for scaling the proximal term in FedProx in accordance with the recommendations provided in (Li et al., 2020b).

### D.1 EXPERIMENTAL SETUP FOR TOY PROBLEMS

**ZDT bi-objective problems** (Zitzler et al., 2000). It contains a class of benchmark problems commonly used to evaluate optimization algorithms, particularly those designed for multi-objective optimization. These problems involve optimizing two conflicting objectives simultaneously. We specifically employ ZDT-1, ZDT-2, and ZDT-3 to evaluate the performance of algorithms. We use multinomial logistic regression, maintaining a consistent learning rate of 0.005 throughout training, with no application of learning rate decay. This configuration aligns with the established setup presented in Liu et al. (2021b). We run 1,000 epochs for the datasets.

### D.2  EXPERIMENTAL SETUP FOR FEDERATED LEARNING

**Synthetic data** (Li et al., 2020a). This synthetic dataset is specifically designed to provide controlled complexities and diverse scenarios for assessing the performance of algorithms. The synthetic data generation process relies on two hyperparameters, $\rho_1$ and $\rho_2$, which shape the dataset's characteristics: $\rho_1$ controls the heterogeneity among local models used to generate labels on each device. While $\rho_2$ governs the differences in data distribution among devices. Larger $\rho_1$ or $\rho_2$ introduces more heterogeneity. For generated dataset, we conduct our experiments using a train-validation-test split ratio of 6:2:2. We use multinomial logistic regression as the model and run 400 epochs in total. The learning rates are swept from $\{0.005, 0.01, 0.05, 0.1\}$ without decaying throught the training process.

**FEMNIST** (Cohen et al., 2017; Caldas et al., 2018). In addition to the synthetic datasets, we also conduct experiments on the Federated Extended MNIST (FEMNIST) dataset, a widely used real-world dataset in federated learning research (Li et al., 2020b), using multinomial logistic regression. It comprises handwritten digit images from multiple users, encompassing 62 classes, including digits (0-9) and uppercase and lowercase letters (A-Z, a-z). The data is distributed across 205 clients, with each client holding a subset of the digit classes. This distribution simulates a real-world federated learning scenario, prioritizing data privacy and distribution concerns. we employ a convolutional neural network featuring two convolutional layers with ReLU activation, followed by max-pooling. Additionally, a fully connected layer maps the flattened features to 62 output classes. We run 400 epochs for training. Learning rates are swept from $\{0.08, 0.1\}$.

### D.3  EXPERIMENTAL SETUP FOR FAIRNESS-ACCURACY TRADE-OFF

For the following two datasets, we employ multinomial logistic regression as our model, conducting 20 epochs of training and sweeping learning rates from $\{0.08, 0.1\}$. And for the following datasets with the number of objectives $n = 2$, we use the enhanced MosT-E. It extends the existing $n$ objectives to $n'$ by interpolating them with weights drawn from a Dirichlet distribution. We set all shape parameters, $\alpha_1, \ldots, \alpha_n$, to the same value within the range $[0.1, 0.5, 1.0]$. The number of extended objectives is chosen from $[10, 15, 20]$. In practice, we observe that extending the original 2 objectives to 10 yields results similar to those obtained with 20 objectives.

**Synthetic dataset** (Zafar et al., 2017). The synthetic dataset contains 2,000 binary classification data instances. These instances are generated randomly as specified in Zafar et al. (2017). The binary labels for classification are generated using a uniform distribution. There are 2-dimensional nonsensitive features that are generated using two different Gaussian distributions. And 1-dimensional sensitive feature is generated using a Bernoulli distribution.

**UCI German credit risk dataset** (Asuncion & Newman, 2007). This dataset comprises 1,000 entries, each characterized by 20 categorical and symbolic attributes. These attributes are used to classify individuals as either good and bad credit risks. We treat the gender as the sensitive attribute here.

### D.4  EXPERIMENTAL SETUP FOR MULTI-TASK LEARNING

In the context of multi-task learning, we employ the Office-Caltech10 and DomainNet datasets to explore the effectiveness of our approach. We use pre-trained model weight as initialization for these two datasets. Specifically, we use ImageNet-pretrained ResNet-18 (He et al., 2016) and ConvNeXt-tiny (Liu et al., 2022), for Office-Caltech10 and DomainNet, respectively. The number of objectives $n$ varies for two datasets, with $n = 4$ for Office-Caltech10 and $n = 6$ for DomainNet.

**Office-Caltech10 dataset** (Saenko et al., 2010; Griffin et al., 2007). The Office-Caltech10 dataset comprises images from four distinct data sources: Office-31(Saenko et al., 2010) (three data sources) and Caltech-256 (Griffin et al., 2007) (one data source). These sources capture images using different camera devices or in various real environments with diverse backgrounds, representing different objectives.

**DomainNet dataset** (Peng et al., 2019). The DomainNet dataset includes natural images sourced from six distinct data sources: Clipart, Infograph, Painting, Quickdraw, Real, and Sketch. This dataset is characterized by its diversity, covering a wide range of object categories. For our experiments, we

focus on a sub-dataset composed of the top ten most common object categories from the extensive pool of 345 categories within DomainNet, following Li et al. (2021).

# E  MULTI-TASK LEARNING EXPERIMENTS

MosT effortlessly extends its application to multi-task learning by treating each task as an individual objective. In this context, $n$ represents the number of tasks to be addressed. Our experiments involve two real-world datasets, namely Office-Caltech10 and DomainNet, with $n = 4$ and 6 objectives, respectively.

To benchmark the effectiveness of MosT, we compare its performance against three state-of-the-art multi-task learning approaches: MGDA, Linearization-based MOO, and EPO (Mahapatra & Rajan, 2020). While we do not compare with Pareto Multi-task Learning (PMTL) (Lin et al., 2019) due to its inadequate computational efficiency, especially when applied to large-scale real-world datasets.

Table 7: Average accuracy across all tasks (mean and std across 3 runs) on Office-Caltech10 and DomainNet. MosT outperforms the strong baselines.

|  | MGDA | Linearization | EPO | MosT |
|---|---|---|---|---|
| Office-Caltech10 | $80.74_{\pm0.44}$ | $61.26_{\pm0.67}$ | $61.05_{\pm1.09}$ | $82.41_{\pm0.23}$ |
| DomainNet | $65.81_{\pm0.37}$ | $57.15_{\pm0.17}$ | $58.55_{\pm0.37}$ | $67.65_{\pm0.55}$ |

Table 7 presents the average accuracy across all tasks for each algorithm, with mean and standard deviation values calculated over three independent runs. The results demonstrate that MosT consistently outperforms the existing approaches on both datasets. In the Office-Caltech10 dataset, MosT achieves an accuracy of 82.41%, surpassing MGDA, Linearization, and EPO by significant margins. Similarly, on the DomainNet dataset, MosT outperforms the three baseline methods. These results underscore the effectiveness of MosT in multi-task learning scenarios, showcasing its capability to achieve superior performance across diverse objectives.

