# OpenReview forum: "Multi-Objective Multi-Solution Transport"
_ICLR.cc/2024/Conference — Submitted to ICLR 2024_

### Official Review · Reviewer_AiMo · 2023-10-13

**Soundness:** 1 poor
**Presentation:** 3 good
**Contribution:** 2 fair
**Rating:** 3
**Confidence:** 5

**Summary:**

This paper proposed a new algorithm based on bilevel optimization. The inner level is an weighted MGDA and the upper level is OT make perference more alginable.

**Strengths:**

The proposed method is validated both on extensive experiments and theory.

**Weaknesses:**

see questions.

**Questions:**

1. Why we do not use a Pareto set learning (PSL) model, which is now a very mature techniques., (see COSMOS[1], Lin[2]) to solve the considered problems. Since PSL is now believed can run very fast and find the whole PS/PF for fairness problems.

2. The considered simplex is m-1 not m. So you should use the notation $\Delta^{m-1}$/$\Delta^{n-1}$ in the paper.

3. Why considered OT? Can a simple JS/KL/TV divergence also work?

4. You claim EPO can not deal non-smooth and dis-continuous problem. But I think the proposed method also have such issues. When the solution is not smooth. It is actually impossible to apply any gradient-based method (to the best of my knowledge).

5. As far as I have implemented, EPO works perfectly on ZDT1. The main problem EPO is it is slow. However, the EPO you have implemented only find a single solution. I suggest the author to re-implement EPO.

6. According to my understanding, LS can work perfectly on MNIST/FMNIST problem since their PFs are almost convex. So I am skeptical to the results in Table 2.

7. The Eq.4, using a weighted version of MGDA, do not have a direct meaning. It is hard to understand its result of Eq4 since the result of MGDA is unpredictable. So a weighted version of MGDA will not lead to very meaningful solutions (according to my try several month ago).

8. I am a curious about the distribution of final solutions (similar to 7).

9. I am actually concerned about the design of the algorithm. I have to say, I think the algorithm is over-designed. Since MGDA does not have a guarantee to find a specific solution. I think COSMOS may be a better choice for the design in the inner loop.

10. I have to say, many of the baselines are not what the author claims. Many of the baselines, (including 5 EPO) actually have much better performance than the author claims.

11. For many-objective problems, MGDA is too slow since calculating m gradients seems not a main-stream method. In addition, how is the OT implement. As far as I know, sinkhorn alg for solving the OT is still slow.


[1] Pareto Set Learning for Expensive Multi-Objective Optimization.

[2] Scalable pareto front approximation for deep multi-objective learning.

---

> ### Author Response · Authors · 2023-11-19
> **Response to Reviewer AiMo**
>
> Thank you for your review on MosT. We appreciate the opportunity to address your concerns and provide additional insights to enhance the understanding of our work.
>
> ---
>
> > Q1. **Why not use a Pareto set learning (PSL) model?**
>    - We acknowledge the existence of techniques like PSL. However, MosT was designed with a focus on handling problems with a large number of objectives (i.e., $n\ll m$, much more than the number of models), aiming for a comprehensive exploration of the Pareto front. We are open to discussing this further and would appreciate any specific objections or concerns you may have.
>
> > Q2. **Optimal Transport (OT) vs. Divergence Measures:**
>    - The choice of OT over other divergence measures was motivated by its ability to capture structural relationships between objectives, which cannot be provided by divergence measures.
>
> > Q3. **Issues on non-smooth solutions:**
>    - The weights assigned by Optimal Transport (OT) exhibit inherent smoothness properties. The optimization process guided by OT involves finding a transport plan that minimizes the loss while satisfying the marginal constraints. This characteristic ensures that the weights smoothly transition between different solutions, contributing to the overall smoothness of the optimization landscape. As a result, MosT benefits from the inherent smoothness of the OT weights, mitigating non-smooth and discontinuous behavior in the optimization process. This contributes to the algorithm's ability to handle a wide range of objective landscapes, promoting stability and reliability across diverse problem scenarios.
>
> > Q4. **EPO Implementation:**
> - We compare the algorithms using their original implementations and a consistent experimental setup, including the use of reference points and the number of solutions. The performance gap may come from the fact that EPO needs to sample an extensive number of preference vectors (\~100) and SVGD needs diverse initializations (\~50). Comparing their numbers with our limited number of solutions (5 for ZDT problems), these two algorithms cannot work as reported. These results show the efficiency of MosT, which does not rely on diverse sampling to achieve well-distributed solutions over the Pareto fronts.
>
> > Q5. **LS should perform well on MNIST/FMNIST because its PF is convex:**
> - Due to data heterogeneity and the large number of clients/objectives, it is unlikely that the Pareto front of FEMNIST (using the convolutional neural network) is convex.
>
> > Q6. **The intuition behind weighted MGDA (Eq.4):**
>    - By the definition of Pareto optimality, the Pareto optimal solutions identified for weighted objectives remain Pareto optimal for the original unweighted problem if all weights are positive. That being said, the weight vector for each model per step guides the MGDA to pursue solutions at a specific region of the Pareto front, similar to a reference/preference vector. Hence, different weight vectors lead to different solutions.
>    - OT-weighted MGDA leads to complementary and balanced solutions, due to the two marginal constraints in OT, which enforce the balance among the $m$ solutions and a fair coverage over all the $n$ objectives. A detailed explanation is provided in General Response #2. This controlled weighting achieves a delicate balance, uncovering solutions that uniquely contribute to a diverse and complementary Pareto front.
>
>
> > Q7. **Distribution of Final Solutions:**
>    - We have shown the distribution in Figure 1.
>
> > Q8. **Computational Efficiency and OT Implementation:**
>   - Optimal transport (OT) only costs a tiny fraction of the total runtime in MosT. Moreover, when compared to alternatives such as linearization and EPO, MosT remains more efficient overall. For comprehensive details and specific values of the runtime, please refer to General Response #3.
>
> ---
>
> We hope these clarifications address your concerns and contribute to an improved understanding of MosT.
>
> Sincerely,
> Authors of Paper #8707

---

> > ### Comment · Reviewer_AiMo · 2023-11-19
> > **wrt epo**
> >
> > I think EPO does not require the number of input. It can accept a single pref.

---

> > > ### Author Response · Authors · 2023-11-19
> > > **Response to 'wrt epo'**
> > >
> > > Yes, EPO can accept a single preference as input but it cannot address the problem studied in our paper. Because it only targets a specific region of the Pareto front (PF) and cannot produce a well-distributed set of diverse solutions over the whole PF. As demonstrated in PMTL [1], for MTL (multi-task learning) problems, sufficient preference vectors are critical to obtain a diverse set of tradeoffs among multiple objectives.
> > >
> > > To empirically verify this, we compare the performance of EPO with different number of preference vectors, i.e., $m \in \\{1,5,10,20,30,40,50\\}$. The results (Hypervolumes) on ZDT-3 are as follows:
> > >
> > > |  m  | 1    | 5    | 10   | 20   | 30   | 40   | 50   |
> > > |-----|------|------|------|------|------|------|------|
> > > | EPO | 1.55 | 4.53 | 4.92 | 5.71 | 6.07 | 6.09 | 6.09 |
> > >
> > >
> > > It shows that **EPO performs poorly when using one single preference vector**. Its performance can be improved by increasing the number of preference vectors, which improves the diversity of the trade-offs by solutions. However, **even with 50 preference vectors, EPO's performance still lags behind MosT, which achieves a hypervolume of 6.36 using only 5 solutions.** This indicates the quality (e.g., diversity and achieved objective values) of preference vectors are more important than their quantity. It underscores the significance of optimizing the guidance weights ($\Gamma$ achieved by OT in MosT, which can be understood as preference vectors in MosT) for achieving diverse high-quality trade-offs on PF.
> > >
> > > [1] Lin X, Zhen H L, Li Z, et al. Pareto multi-task learning. Advances in neural information processing systems, 2019, 32.

---

> > > > ### Comment · Reviewer_AiMo · 2023-11-20
> > > > **about epo**
> > > >
> > > > Why on zdt1, epo can only find a single solution？ （Figure 6 in the main paper）. I have just run this experiment several dates among. EPO work perfectly on ZDT1, do you use the same preference?

---

> > > > > ### Author Response · Authors · 2023-11-20
> > > > > **Response to "about epo"**
> > > > >
> > > > > In Figure 6, all methods only use 5 preference vectors and only find at most 5 solutions.
> > > > >
> > > > > As we mentioned in our response above, if we increase the number of reference vectors for EPO from 5 to >40, its solutions will be more diverse than those in Figure 6, and the HV will be improved. But MosT only needs m=5 solutions to outperform the HV achieved by EPO when m>40.

---

> > > > > > ### Comment · Reviewer_AiMo · 2023-11-20
> > > > > > **a tricky thing about epo**
> > > > > >
> > > > > > a tricky thing about epo is that, it is the exact solution w.r.t. the inverse of the preferences $1/\lamba_1, ..., 1/\lamba_m$. What the current chosen of preferences. A fair comparasion should be, the uniform preference distribution in this inverse preference space.

---

> > ### Comment · Reviewer_AiMo · 2023-11-19
> > **what do you mean struture information**
> >
> > does it help for the final performance?

---

> ### Comment · Reviewer_AiMo · 2023-11-20
> **Another missing work.**
>
> Another missing work is "Multi-Objective Deep Learning with Adaptive Reference Vectors", which use a similar bi-level optimization method by maximizing the hv.
> Since this paper seems that directly optimize the hv. This method should have better hv than Most. Since Most does not directly optimize hv.

---

> ### Author Response · Authors · 2023-11-20
> **Response to "Another missing work"**
>
> Thanks for pointing out the work! We will look into it and discuss it in our paper. Since we cannot find the source code of the paper online, it might be hard to provide an empirical comparison with it within a short time.
>
> That being said, most HV optimization methods only work for small-$n$ cases ($n$=2 or 3 objectives). They usually CANNOT scale to **large-$n$ and $n\gg m$ cases** with hundreds to thousands of objectives, which are the main focus of this paper and the gap we aim to bridge. The computation of HV and its gradient may also suffer from the curse of dimensionality when $n$ is large.
>
> **In Table 4 of the paper you mentioned, they already claimed that their method is hard to scale to merely $n=15$ objectives**. Quoted from Table 4's caption, "GMOOAR-HV cannot be run on 15 tasks as it takes more than a month on our machine." In contrast, in Section 5.3 of our paper, **MosT can easily scale to $n>200$ objectives**.

---

> > ### Comment · Reviewer_AiMo · 2023-11-20
> > **current concerns**
> >
> > My current concern is mainly about the experiments.
> > (1) When the number of  objective n is small. I think, it is actually pretty hard to improve the previous methods. Chen Wei Yu's NeurIPS work, considering the hv maximization should have the best hv, though hv may still not a satisfied indicator.
> >
> > (2) I agree with the author, that, the many objective issue is very important. However, the results shown in Figure 2(c) outperform other methods by such a great margin. I think this is impossible. Other methods may not get enough diverse objectives, but it seems that other solution is dominated by the proposed method on all objectives. The results seems too optimistic.
> >
> > What is the core reason, the proposed method win such a lot on Figure (a)? The considered different type MGDA or OT?

---

> ### Author Response · Authors · 2023-11-20
> **Response to "a tricky thing about epo" and "current concerns"**
>
> Q1. What are the current chosen preferences?
>
> The preference vectors used for EPO are generated using the exact pipeline as outlined in the official implementation of EPO (https://github.com/dbmptr/EPOSearch/blob/master/mtc/epo_train.py). That is, we follow the same process described as the "tricky way" in your review. We are committed to providing the implementation details of MosT, as well as all baseline methods, upon publication.
>
> Q2. It seems that other solution is dominated by the proposed method on all objectives.
>
> In Figure 2(c), we are not displaying the performance gap for each objective separately but rather for groups of objectives. So, the figure does not suggest that MosT dominates other solutions on all objectives. What it illustrates is that MosT outperforms the baselines by a larger margin for clients with worse performance. This underscores MosT's role in promoting fairness in federated learning by delivering superior results for those clients facing greater challenges.
>
> Q3. Why does MosT win a lot in federated learning?
>
> This is because we keep optimizing the assignment of solutions (models) to objectives (clients) in the training stage (by OT), while other methods do not optimize the assignments at all.

---

> > ### Comment · Reviewer_AiMo · 2023-11-20
> >
> > I do believe EPO can lead to perfect result on zdt1. I will conduct this experiment for you tomorrow.

---

### Official Review · Reviewer_EZcQ · 2023-10-31

**Soundness:** 2 fair
**Presentation:** 2 fair
**Contribution:** 2 fair
**Rating:** 5
**Confidence:** 4

**Summary:**

This work proposes a novel Multi-objective multi-solution Transport (MosT) method to find a small set of diverse Pareto solutions for problems with a large number of objectives. MosT is formulated as a bi-level optimization problem, where the upper-level problem is to determine different objective weights for each solution via optimal transport (OT) based on their current performance, while the lower-level problem is to find a Pareto solution for each subproblem with weighted objectives. MosT can also be generalized to tackle multi-objective optimization problems with few objectives via random objective interpolation.

This work also provides theoretical analysis to prove MosT can find a set of Pareto solutions via solving the bi-level optimization problem. Experiments show MosT can achieve promising performance on different synthetic and application problems.

**Strengths:**

+ This work is well-organized and easy to follow.

+ The *small solution set for a large number of objectives* setting is important for many real-world applications. This work is a timely contribution to an interesting yet under-explored research direction.

+ The proposed MosT method can achieve promising performance on different synthetic and application problems.

**Weaknesses:**

I enjoy reading this paper but also have many major concerns on the proposed MosT method, which makes it hard for me to vote for acceptance. My current rating is more like a weak reject (4).

**1. Motivation**

Some crucial design choices for MosT are not well motivated, and more clear discussions are needed.

- **Weighted Objectives:** The key of MosT is to assign different weights to the objectives for different solutions. What is the relation of the Pareto sets for the problems with weighted objectives (1) with the Pareto set of the original unweighted problem? In addition, It is also unclear what makes the weighted problems good for diversity. On page 4, it mentions that the weighted problem can move a small gradient from the origin in Eq. (5). But the whole picture of how the (optimal) weighted problem can lead to a set of "different but complementary and balanced" solutions that best cover all objectives is not clear to me.

- **Optimal Weights and OT:** If there is a set of optimal weights that can lead to a set of optimally distributed solutions, what properties such a set of weights should have? Is it unique? Why such optimal weights (if they exist) can be found by OT (e.g., eq.(2)) is also not clearly discussed.

- **MGDA for Weighted Problems:** MosT uses MGDA to find a Pareto solution for each problem with different weighted objectives. However, MGDA can find any Pareto stationary solutions and the location is not controllable. For a non-trivial problem, the Pareto set could be a large $n-1$ dimensional manifold with a large $n$ for each weighted problem, where MGDA can lead to any possible solution on the manifold. Why the (uncontrollable) solutions found by MGDA for each weighted problem (1) can be guaranteed to be complementary and evenly distributed?

**2. Gap between MosT and the Metrics**

There are different and not unique metrics to measure the quality of the obtained solution set. It is unclear why the solution set found by MosT could be optimal for a given criteria (e.g., hypervolume).

- **Hypervolume for Few Objectives $(n << m)$:** For problem with few objectives (e.g., 2-3), if the goal is to maximize the hypervolume, why not directly use gradient-based hypervolume maximization to find the solution set [1,2]?

- **Large Number of Objectives $(n >> m)$:** For problems with a large number of objectives, this work let each objective pick the best solution out of all $m$ obtained solution, and then calculate the overall performance. In this case, it seems that, at the end of optimization, the objectives covered by different optimal solutions should be mutually exclusive. For example, for a problem with 300 objectives and 3 solutions, solution 1 covers 95 objectives, solution 2 covers 103 objectives, and solution 3 covers 102 objectives, and hence each objective is covered by a single solution. In this case, the optimal solution should not cover other conflicted objectives that it does not work the best, since it might lower the solution's performance on the currently covered objectives.  Therefore, it is unclear why "During later stages, ..., every objective has to be covered by sufficient models (solutions)".

In addition, the final metric we care about is the mean(std) of average performance across all objectives (e.g., Table 2). Once the groups of objectives covered by different solutions are determined, why not just use simple uniform linear scalarization to optimize all objectives for each group?

**3. Time Complexity**

- MosT formulate the original multi-objective optimization as iterative bi-level optimization, which involves solving a OT problem and $m$ MGDA problems at each iteration. What is the time complexity of MosT? Will the OT solver has high computational overhead?

- For MGDA with large $n$ (e.g., 205), even with the Frank-Wolfe algorithm, will the run time be very long for solving eq. (5)? Will it be very hard to find a valid gradient direction to optimize all 205 objectives at the same time?

**4. Extension to Few Objective Case ($n << m$)**

For problems with few objectives (e.g., 2 or 3), this work uses $n^{\prime}$ dense interpolations among original objectives to create a large number of objectives, and then applies MosT to find a set of diverse solutions.

- If $n^{\prime}$ is large, will this approach lead to high computational overhead for problems with few objectives?

- The weights of interpolation are uniformly drawn from a Dirichlet distribution over the simplex. However, uniform weights usually do not represent uniform solutions on the Pareto front. If $n^{\prime}$ is small, will it hurt the performance?

**5. Experiment**

- **Toy Problems:** To my understanding, the algorithms considered in Section 5.1 are all for unconstrained multi-objective optimization problems. However, the ZDT problems have box constraints ($\theta \in [0,1]^d$), and their Pareto sets are all on the boundary (the constraints are activated). Will these algorithm's performances be significantly affected by the constraints? In addition, according to [3], EPO (if with a suitable reference point) and SVGD have much better performance on ZDT1-3. What is the reason for the poorer performance reported in this paper?

- **More Problem with Large Number of Objectives:** The main motivation of this work is to tackle the *small solution set for a large number of objectives* problem. However, the experiment only has a single real-world application with many objectives on the federated learning application with FEMNIST. The other experiments are on synthetic problems or problems with few objectives. It is important to know MosT's performance on more realistic application problems with a large number of objectives, especially those other than federated learning.

- **Runtime Comparison:** Please report the runtime for different algorithms on all experiments.

**Questions:**

- Please address the concerns raised in the above weaknesses.

- When citing multiple papers, it is better to put them in chronological order.

**Reference**

[1] Hypervolume indicator gradient ascent multi-objective optimization. International Conference on Evolutionary Multi-Criterion Optimization 2017.

[2] Multi-Objective Learning to Predict Pareto Fronts Using Hypervolume Maximization. arXiv:2102.04523.

[3] Profiling Pareto Front With Multi-Objective Stein Variational Gradient Descent. NeurIPS 2021.

---

> ### Author Response · Authors · 2023-11-19
> **Response to Reviewer EZcQ (part 1)**
>
> ### 1. Motivation
> > Q1.1  **What is the relation of the Pareto sets for the problems with weighted objectives (1) with the Pareto set of the original unweighted problem? How the (optimal) weighted problem can lead to a set of "different but complementary and balanced" solutions that best cover all objectives?**
>
> By the definition of Pareto optimality, the Pareto optimal solutions identified for weighted objectives remain Pareto optimal for the original unweighted problem if all weights are positive. That being said, the weight vector for each model per step guides the MGDA to pursue solutions at a specific region of the Pareto front, similar to a reference/preference vector. Hence, different weight vectors lead to different solutions.
>
>
> The complementary and balanced solutions are enforced through the weights achieved by OT and they are mainly due to the two marginal constraints in OT, which enforces the balance among the $m$ solutions and a fair coverage over all the $n$ objectives.
> This controlled weighting achieves a delicate balance, uncovering solutions that uniquely contribute to a diverse and complementary Pareto front.
>
> > Q1.2  **If there is a set of optimal weights that can lead to a set of optimally distributed solutions, what properties such a set of weights should have? Is it unique? Why such optimal weights (if they exist) can be found by OT (e.g., eq.(2)) is also not clearly discussed.**
>
> Optimal weights in MosT are expected to achieve diverse and complementary assignments from solutions to objectives. The solution to entropic optimal transport, as employed in MosT, is unique [1]. OT seeks a transportation plan that minimizes loss (complementary) while meeting the marginal constraint (diversity) [2]. This offers a clear connection between the principles of OT and the optimal weights needed in MosT.
>
> > Q1.3  **Why the (uncontrollable) solutions found by MGDA for each weighted problem (1) can be guaranteed to be complementary and evenly distributed?**
>
> MosT combines MGDA with strategic weight assignment for a balanced exploration of the Pareto front. While MGDA itself may not guarantee the location of solutions, the reweighted objective (with weights 'controlled' by OT) ensures that the algorithm converges to solutions that cover different regions of the Pareto front. The weights assigned to objectives are derived through Optimal Transport (OT), introducing a controlled and strategic weighting mechanism. This weighting process ensures that the solutions found by MGDA are not arbitrary but guided by a deliberate assignment of weights.

---

> ### Author Response · Authors · 2023-11-19
> **Response to Reviewer EZcQ (part 2)**
>
> ### 2. Gap between MosT and the Metrics
>
>
> > Q2.1  **Why not directly use gradient-based hypervolume maximization to find the solution set?**
>
> While gradient-based hypervolume maximization is effective for a small number of objectives, MosT is effective in both less-dimensional and high-dimensional scenarios. Combining multi-gradient descent algorithms with optimal transport, MosT navigates high-dimention Pareto front efficiently.
>
>
> Additional, we want to highlight challenges with direct hypervolume maximization [3], facing issues such as dependence on reference points and susceptibility to bad local minima [4,5].
>
> > Q2.2  **It is unclear why "During later stages, ..., every objective has to be covered by sufficient models (solutions)."**
>
> During later stages, the uniformity of objective marginals, coupled with performance-based solution marginals, enables each objective to select its best-performing solution(s).
>
> We acknowledge that objectives may be covered by different optimal solutions, raising concerns about potential mutual exclusivity. However, MosT aims to harmonize these contributions rather than enforce exclusivity. MosT strategically balances marginal distributions in later stages as described. This design ensures that even if a solution excels on specific objectives, the overall objective coverage remains diverse and robust.
>
> > Q2.3 **In addition, the final metric we care about is the mean(std) of average performance across all objectives (e.g., Table 2). Once the groups of objectives covered by different solutions are determined, why not just use simple uniform linear scalarization to optimize all objectives for each group?**
>
>
> Our main interest is in achieving diverse and complementary assignments from solutions to objectives, not just in terms of average accuracy. We want to make sure our solutions cover various objectives effectively.
>
> Note that, even within the same group you mentioned, objectives may exhibit varying characteristics and trade-offs. Consequently, a one-size-fits-all approach, such as simple uniform linear scalarization, might not capture the intricacies of each objective grouping. In addition, such alternative approaches require additional design, such as stopping criteria, which can increase the complexity of the algorithm. In contrast, MGDA tailors the optimization process to the specific characteristics of each group. This fine-grained optimization approach ensures that each objective's contribution is considered with the attention it deserves.
>
>
> ### 3. Time Complexity
>
> We acknowledge the importance of this aspect and have provided a detailed analysis in General Response #3. In summary, optimal transport constitutes only a small fraction of the total runtime in MosT. Moreover, when compared to alternatives such as linearization and EPO, MosT remains more efficient overall. For comprehensive details and specific values, please refer to General Response #3.
>
> ### 4. Extension to Few Objective Case (n<<m)
>
> We empirically show that a relatively small $n'$ for interpolations of a few (i.e., $n$) original objectives is sufficient to achieve adequate performance. In the accuracy-fairness tradeoff ($n=2$), we conducted experiments sweeping over $n'\in \{10, 20, 30, 40, 50\}$. Our results indicate that utilizing $n' = 10$ yields comparable performance to those larger $n'$, as clarified in Appendix D.3. Consequently, we adopted the smaller $n' = 10$ in our experiments, obtaining the final results reported in Table 3.

---

> ### Author Response · Authors · 2023-11-19
> **Response to Reviewer EZcQ (part 3)**
>
> ### 5. Experiment
>
> > Q5.1 **Constraints in ZDT Problems and performance of EPO and SVGD:**
>    - All algorithms in the experiments, including MosT, are originally designed to handle unconstrained multi-objective optimization problems. To enforce the constraints in the ZDT problem, all algorithms perform projected gradient descent, i.e., by clipping the variables back to [0,1] in every step if necessary. This operation ensures the fairness of the comparison and the constraints.
>
>
>    - We compare algorithms with their original implementations and consistent settings, including the same reference point and the same number of solutions. The performance gap may come from the fact that EPO needs to sample an extensive number of preference vectors (\~100) and SVGD needs diverse initializations (\~50). Comparing their numbers with our limited number of solutions (5 for ZDT problems), these two algorithms cannot work as reported. These results show the efficiency of MosT, which does not rely on diverse sampling to achieve well-distributed solutions over the Pareto fronts.
>
> > Q5.2 **Realistic Application Problems with a Large Number of Objectives:**
>    - **Expansion to Multi-Task Learning Scenarios:** We appreciate your suggestion to include more realistic application scenarios with a higher number of objectives. In response, we have expanded our experiments to include multi-task learning scenarios with two real datasets (n = 4 and 6, resp) and showed promising results. This addition aims to provide a more diverse and comprehensive evaluation of MosT's performance across different application domains. Detailed results and analysis are available in Appendix E and General Response #4.1.
>
> > Q5.3 **Runtime Comparison:**
>
>    - In response to your feedback, we have included detailed runtime metrics for MosT and other algorithms over two real datasets, detailed in General Response #3. The results show that MosT is more efficient compared to linearization and EPO, and comparable to MGDA.
>
> Sincerely,
> Authors of Paper #8707
>
> ---
>
> References:\
> [1] Xie Y, Wang X, Wang R, et al. A fast proximal point method for computing exact wasserstein distance. Uncertainty in artificial intelligence. PMLR, 2020: 433-453.\
> [2] Cuturi M. Sinkhorn distances: Lightspeed computation of optimal transport. Advances in neural information processing systems, 2013, 26.\
> [3] Liu X, Tong X, Liu Q. Profiling pareto front with multi-objective stein variational gradient descent. Advances in Neural Information Processing Systems, 2021, 34: 14721-14733.\
> [4] Wang H, Deutz A, Bäck T, et al. Hypervolume indicator gradient ascent multi-objective optimization. Evolutionary Multi-Criterion Optimization: 9th International Conference, EMO 2017, Münster, Germany, March 19-22, 2017, Proceedings 9. Springer International Publishing, 2017: 654-669.\
> [5] Deist T M, Grewal M, Dankers F J W M, et al. Multi-objective learning to predict pareto fronts using hypervolume maximization. arXiv preprint arXiv:2102.04523, 2021.

---

> ### Comment · Reviewer_EZcQ · 2023-11-21
> **Follow-up Comment**
>
> Thank you for your detailed response and additional experiments. However, many of my concerns remain.
>
> **1. Motivation:**  Many claims are not solid and not well-supported, such as "the weight vector for each model per step guides the MGDA to pursue solutions at a specific region of the Pareto front, similar to a reference/preference vector" and "this weighting process ensures that the solutions found by MGDA are not arbitrary but guided by a deliberate assignment of weights".
>
> In MosT, the MGDAs can reach any Pareto optimal solution of each weighted problem, which is exactly the same Pareto optimal set of the original problem. In other words, the current theoretical guarantee of MosT is not stronger than the original MGDA. The reason why MosT can be ensured to find a set of diverse Pareto solutions does not have proper theoretical support.
>
> **2. Metrics:** My main concern here is to clearly show why the solution set found by MosT can outperform other algorithms on different criteria for both the (n<<m) and (n>>m) cases with *theoretical analysis*.
>
>
> **3. Time Complexity:** It is counter-intuitive to see linear scalarization require more runtime than MGDA. What is the computational overhead of linear scalarization over MGDA? In addition, since the main motivation for MosT is to address the problem with a much larger number of objectives (n >> m), the runtime comparison under this setting (e.g., with n = 205) is much more important.
>
> **5.1 Toy Problems:** An immediate follow-up question is why compare all unconstrained multi-objective optimization algorithms on a set of constrained problems?
>
> In addition, to my understanding, EPO is an exact preference-based optimization method, and the quality of each solution is agnostic to the number of preference vectors. Even only given a single preference vector, under a set of reasonable assumptions, EPO can also successfully find the Pareto solution with the specific trade-off. Therefore, EPO should work fine with 5 given preference vectors. The claim "EPO needs to sample an extensive number of preference vectors (~100)" is questionable.
>
> **5.2 More Problem with Large Number of Objectives:** The main motivation for MosT is to handle the problem with a large number of objectives (e.g., m << n = 200). The added experiments with a small number of objectives (e.g., n = 4 and 6) are still far from the main motivation. Is it very hard to find a practical application for the n >> m setting?
>
> In addition, recent works on MTL and multi-domain learning have shown that simple linear scalarization with proper regularization can perform comparably with advanced methods such as MGDA [4,5,6]. The linear scalarization is also fast and will not suffer from a large computational overhead for problems with many tasks (e.g., a large n) as considered in this work. Why is the performance of linear scalarization much worse than MGDA in the new experiments with a large gap?
>
> [1] In Defense of the Unitary Scalarization for Deep Multi-Task Learning. NeurIPS 2022.
>
> [2] Do current multi-task optimization methods in deep learning even help. NeurIPS 2022.
>
> [3] Scalarization for Multi-Task and  Multi-Domain Learning at Scale. NeurIPS 2023.
>
> **5.3 Runtime Comparison:** Please report the runtime for the n >> m setting (e.g., n = 200).

---

### Official Review · Reviewer_SZEd · 2023-11-02

**Soundness:** 3 good
**Presentation:** 3 good
**Contribution:** 3 good
**Rating:** 5
**Confidence:** 3

**Summary:**

This paper proposes a method to generate $m$ solutions over the Pareto Set of a MOO to maximize diversity among the solutions, especially in the case where the number $n$ of objectives are much larger than $m$.

To enforce diverse solutions, this paper uses an Optimal Transport(OT)-based approach to learn the weights of different objectives towards evaluating different solutions such that the solutions are diverse and then uses MGDA to solve the MOO.

The advantages of the method have been illustrated over several experiments involving toy and real datasets.

**Strengths:**

1. The main strength of the method is summarized nicely after eq (6) in the paper. Incorporating the optimal transport between uniform distribution over $1$ to $n$, and $1$ to $m$ effectively reweights the objectives before running vanilla MGDA. Because of OT, these weights $\Gamma_{i,j}$ are larger where $L_i(\theta_j)$ are small, that is, when two objectives $i$, and $j$ are similar ($L_i(\theta_j)$ small), large $\Gamma_{i,j}$ decreases the influence of the gradients of such objectives over MGDA compared to objectives which are different from each other. In the absence of $\Gamma_{i,j}$ the solution of MGDA gets biased towards the objectives with small gradients and may lead to smaller diversity.

2. This nice intuition works well over several applications. The effect of incorporating OT is nicely demonstrated in Figure 4 which shows the diversity in the selected objectives.

**Weaknesses:**

The main weakness of the paper is the lack of theoretical support. **While the main claim of the paper is that it learns diverse solutions over Pareto set when the number of objectives is large, Theorem 1 and 2 only show convergence of the methods to any $m$ Pareto solutions which sheds no light on the diversity of the solutions.** The authors acknowledge this in the line just before Theorem 2.

The convergence results follow straightforwardly from existing literature as it just combines the proof of convergence of IPOT and MGDA. Moreover, it just characterizes the complexity of the outer loop whereas the inner loop (especially IPOT) can be quite computationally expensive.

Without proper theoretical justification, the claimed advantages of MosT, diverse solutions, and computationally cheaper, over other methods seem weak.

**Questions:**

1. Why is $\epsilon$ needed in Algorithm 1?

2. I would expect the hypervolume of the solutions to be higher if one runs MGDA with diverse weight vectors $m$ times instead of random seeds. Could you compare MosT with such a version of MGDA?

3. While comparing to previous work it is stated in Page 2 that the number of preference vectors required to profile the Pareto set can be exponential in the problem parameters which might be discouraging when the number of objectives $n$ is large. When $m$ and/or $n$ are large, IPOT (Line 4 in Algorithm 1) needs to be solved for high dimensions which can still be computationally large. How does this paper compare with the previous literature on this aspect?

Minor Points (do not affect my score):
1. Typo in Page 6: MDGA --> MGDA
2. The full form of the abbreviations should be written where they are first introduced.
3. ``which gives higher priority to models selecting objectives at
the earlier stages and then transits to a higher priority of objectives selecting the best models." - Why is this desirable?

---

> ### Author Response · Authors · 2023-11-18
> **Response to Reviewer SZEd**
>
> Thanks to Reviewer SZEd for the insightful comments, and we provide our response below.
>
> > **Q1. Why is $\epsilon$ needed in Algorithm 1?**
>
>
> Setting $\epsilon$ to be a small non-zero constant guarantees that any Pareto solutions for the re-weighted objective in Eq.(5) are also Pareto solutions for the original unweighted problem. Empirically, we found that it also provides numerical stability during the optimization process. Its effectiveness is also supported by our convergence, which incorporates a non-zero $\epsilon$. We will add more details in the revision.
>
>
> > **Q2. Comparison with MGDA using diverse weight vectors $m$ times.**
>
> Following your suggestion, we conduct an ablation study in the context of federated learning with $n\gg m$, in order to compare MosT and MGDA with $m$ different weight vectors for $n$ objectives. Our ablation study shows MosT with OT-generated weights achieves higher accuracy averaged across all objectives than the MGDA alternative. This indicates the importance of optimal transport in simultaneously balancing multiple solutions and multiple objectives. Comprehensive details and results are available in Appendix C.3 and General Response #4.2.
>
> > **Q3. Comparison with Previous Literature on Computational Aspects.**
>
> In our method, the number of solution $m$ is a predefined constant, while MosT aims to maximize their usage and fully exploit their capacity by the optimal transport assignment to the $n$ objectives. This leads to better model efficiency, for example, when $n\gg m$. In contrast, previous work usually requires the number of models $m$ to grow linearly or exponentially with $n$, without explicit optimization of their matching. Hence, our design aligns well with the practical need of computational efficiency and model efficiency. It also leads to scalability and computational tractability in scenarios with a large number of objectives ($n\gg m$). In General Response #3, we provide the end-to-end running time of our method and the baselines under the same $m$, showing that MosT is able to result in better solutions more quickly.
>
> ---
>
> We hope these responses address your concerns and provide further clarity on the contributions of our work.
>
> Sincerely,
> Authors of Paper #8707

---

### Official Review · Reviewer_bAug · 2023-11-09

**Soundness:** 1 poor
**Presentation:** 2 fair
**Contribution:** 1 poor
**Rating:** 1
**Confidence:** 5

**Summary:**

The authors introduced a multi-objective multi-solution transport approach to optimizing multiple objectives using multiple solutions, which aims to achieve diverse trade-offs among objectives by treating each solution as a domain expert. The authors stated that the approach addresses cases where the number of objectives greatly exceeds the number of solutions and demonstrates superior performance in applications like federated learning, fairness-accuracy trade-offs, and standard multi-objective optimization benchmarks, providing high-quality, diverse solutions that cover the entire Pareto frontier. Moreover, the authors stated that the approach “aims to find m Pareto solutions (models) that achieve diverse trade-offs among n optimization objectives”.

**Strengths:**

NA

**Weaknesses:**

The literature review does not provide a cohesive and structured presentation, and fails to accurately acknowledge established approaches and terminology within Evolutionary Computation (EC). This indicates a potential oversight in acknowledging state-of-the-art methods in EC such as the author do not properly reference existing work on this topic from EC. The approach lacks novelty and has been previously explored in the literature. The description of the approach and the literature review is not presented in a clear and understandable manner. The authors do not provide any new insights into the method, key experimental setup is missing (see section 5.1 Experimental setting). The solutions produced in Section 5 “MOST APPLICATIONS” are not clearly described and it is not clear that there is any significance to the results taking into account the benchmarks are insufficiently small and inadequate for comprehensive evaluation. The paper's contributions lack substantial significance and originality, and primarily represent incremental progress.

**Questions:**

How does this approach contribute to achieving diverse trade-offs among objectives? Can you provide more details about the theoretical foundation?
Can you provide a more detailed comparison between the MosT approach and existing state-of-the-art methods in multi-objective optimization?

---

> ### Author Response · Authors · 2023-11-17
> **Response to Reviewer bAug**
>
> We express gratitude to Reviewer bAug for the evaluation of our submission. While we respect your perspective, we aim to address concerns and provide clarifications.
>
> ---
>
> > Q1. **Literature Review:**
>    - Our literature review in Section 2 did include discussions on Evolutionary Computation (EC) methods. We also compared MosT with SVGD [1], a recent approach that evolutionarily updates particles towards the Pareto front. Specifically, EC methods can be inefficient in practical MOO problems due to the absence of gradient information [1-3] and the poor exploration in high-dimensional spaces (when the number of objectives is large). This distinction positions our approach, which leverages gradient-based techniques, as a more suitable and efficient alternative.
>
>    - **Question: Could you please specify which literature is missing?**
>
> > Q2. **Novelty and Significance:**
>    - Our primary contributions and novelty: (1) the novel task and problem formulation of multi-objective multi-solution transport; (2) addressing a unexplored class of challenging problems when $n\gg m$ (but also appliable to cases when $n\ll m$); (3) intuitive and effective MosT algorithm; (4) thorough empirical evaluation on a diverse set of machine learning problems. Specifically, the marginal constraints in Optimal Transport (OT) are essential to achieve a balanced solution-objective matching, leading to a diverse and complementary ensemble of solutions.
>
>
>   - **Question: Could you please specify which literature has explored our method?**
>
>
> > Q3. **Experimental Setup:**
>    - The details of the experimental setup are introduced in Section 5.1, and additional information is provided in Appendix D for a more comprehensive understanding.
>
>    - **Question: Could you please specify which part of the experimental setup is considered missing?**
>
> > Q4. **Results Significance:**
>
>    MosT proves its application across federated learning, fairness-accuracy trade-offs, and multi-task learning (new to rebuttal) with multiple real datasets and also synthetic datasets.
>    - Our dataset selection aligns with established practices from previous papers in federated learning [4] and multi-objective optimization [1].
>    - In the new version, we have expanded our experiments to real-world datasets for multi-task learning, providing a more comprehensive evaluation for $2\leq n\ll m$ cases. Detailed results and analysis are available in Appendix E. We will discuss this in the manuscript to underscore the significance of our findings.
>
> In conclusion, we appreciate the opportunity to address these concerns and are committed to enhancing the clarity and completeness of our manuscript based on your valuable feedback.
>
> Sincerely,
> Authors of Paper #8707
>
> ---
>
> References:\
> [1] Liu X, Tong X, Liu Q. Profiling pareto front with multi-objective stein variational gradient descent. Advances in Neural Information Processing Systems, 2021, 34: 14721-14733.\
> [2] Mahapatra D, Rajan V. Multi-task learning with user preferences: Gradient descent with controlled ascent in pareto optimization. International Conference on Machine Learning. PMLR, 2020: 6597-6607.\
> [3] Momma M, Dong C, Liu J. A multi-objective/multi-task learning framework induced by pareto stationarity. International Conference on Machine Learning. PMLR, 2022: 15895-15907.\
> [4] Li T, Sahu A K, Zaheer M, et al. Federated optimization in heterogeneous networks[J]. Proceedings of Machine learning and systems, 2020, 2: 429-450.

---

### Official Review · Reviewer_YDRF · 2023-11-10

**Soundness:** 3 good
**Presentation:** 3 good
**Contribution:** 3 good
**Rating:** 8
**Confidence:** 4

**Summary:**

This articled presented a multi-objective multi-solution transport framework aiming to find the solutions that achieve diverse trade-offs among n optimization.

**Strengths:**

This article explored the feasibility of exploring a high dimensional Pareto frontier where there needs significant contribution is needed.
The authors framework had theoretically converges to a number of solution by optimizing the objectives and optimal transport.
Applied the framework to some of the ML problems such as federated learning, fairness-accuracy trade-offs, some other multi-objective optimization benchmark problems.
Empirical articulation of convergence analysis

**Weaknesses:**

Improvements and contributions towards federated learning would ensemble this article to a different altitude

**Questions:**

Interesting analysis on n<<m in section 5.4. But the number of objectives is set as 2. Are there experiments conducted with more than 2 objectives? Did the solution converge when n increased to 3 or 5?
What type of solver used to run needs to be detailed out
Keen to understand federated learning in detail, how does the clients receive training as it is very diverse and also what's the effect of number of local sample (vi) in such scenario?

---

> ### Author Response · Authors · 2023-11-18
> **Response to Reviewer YDRF**
>
> We extend our sincere thanks to Reviewer YDRF for the insightful evaluation of our submission. We are encouraged by your recognition of the significance of our work in exploring a high-dimensional Pareto frontier. We hope the following replies resolve the concerns raised in your review.
>
> ---
>
> > Q1. **"Are there experiments conducted with more than 2 objectives? Did the solution converge when n increased to 3 or 5?"**
>
> Following your suggestion, we've **expanded our experiments to include multi-task learning scenarios with `n << m` and varying numbers of objectives beyond two ($n=4 \text{ and } 6$)**. Results show that MosT achieves promising performance over baselines, supporting the versatility of our proposed framework as `n` increases. Detailed results and analysis are available in Appendix E of the revised manuscript and also in General Response #4.1.
>
> > Q2. **"What type of solver used to run needs to be detailed out."**
>
> We utilize IPOT [1] as an off-the-shelf OT solver for Eq.(2) and a min-norm solver (based on a Frank-Wolfe algorithm) used by the vanilla MGDA [2] for optimizing Eq.(1), as mentioned in Section 3. We found that these solvers are effective in handling the complexities of our proposed optimization tasks.
>
> > Q3. **(also mentioned in `[Weakness]`) "Keen to understand federated learning in detail, how do clients receive training, and what's the effect of the number of local samples (vi)?"**
>
> Thanks for your question. Clients perform training by running local SGD updates, similar to standard federated optimization methods.
>
> In response to your suggestion, we conducted experiments that involved varying numbers of local samples per client—specifically, 50, 75, and 100—utilizing the synthetic federated learning dataset. Experiments consistently demonstrated MosT's superior performance compared to all baseline methods, as shown in the table below. Notably, **MosT exhibited greater efficacy, particularly with smaller local sample sizes**, outperforming the best baseline (MGDA).
>
> |     |  MGDA | Linearization | FedAvg | FedProx | FedMGDA+ | MosT  | over MGDA (%) |
> |:---:|:-----:|:-------------:|:------:|:-------:|----------|-------|-------|
> |  50 | 82.44 |     81.56     |  81.63 |  81.85  | 79.33    | 85.33 | 3.51% |
> |  75 | 83.89 |     81.33     |  82.56 |  82.67  | 77.44    | 86.56 | 3.18% |
> | 100 | 87.50 |     85.00     |  84.11 |  84.56  | 84.08    | 89.06 | 1.78% |
>
>
> Sincerely,
> Authors of Paper #8707
>
> ---
>
> References:
>
> [1] Xie Y, Wang X, Wang R, et al. A fast proximal point method for computing the exact Wasserstein distance. Uncertainty in artificial intelligence. PMLR, 2020: 433-453. \
> [2] Désidéri J A. Multiple-gradient descent algorithm (MGDA) for multiobjective optimization. Comptes Rendus Mathematique, 2012, 350(5-6): 313-318.

---

### Author Response · Authors · 2023-11-18
**General Response (part 2)**

**#4 New experiments**

We provide additional explanations and new experiments per the reviewers' comments.

***#4.1 Experiments on multi-task learning with two real datasets***

**We have expanded our experiments to multi-task learning with the number of objectives beyond two ($2< n \ll m$)**, specifically, $n=4$ (on Office-Caltech10) and $n=6$ (on DomainNet). The average accuracies, reported in the table below, showcase MosT's superior performance compared with baseline methods. This underscores the versatility of our proposed framework as $n$ increases. Detailed results and analysis are available in Appendix E of the revised manuscript.

|                  |  MGDA | Linearization | EPO   | MosT      |
|:----------------:|:-----:|:-------------:|-------|-----------|
| Office-Caltech10 | 80.74 |     61.26     | 61.05 | **82.41** |
|     DomainNet    | 65.81 |     57.15     | 58.55 | **67.65** |


***#4.2 MGDA using diverse weight vectors m times (i.e., effectiveness of OT in MosT)***


In response to Reviewer SZEd's comment, we have conducted an ablation study to compare the performance of MosT versus MGDA with $m$ random weight vectors (`w-MGDA`). This comparison was performed on three synthetic federated learning datasets, and the detailed results are provided in Appendix C.3. The results are reported in the table below. (The `MosT (L)` baseline will be introduced in the next response, which can be ignored for now.)

|                | MGDA  | w-MGDA | MosT (L) | MosT      |
|----------------|-------|--------|----------|-----------|
| Syn (0.0, 0.0) | 77.22 | 77.96  | 79.89    | **83.09** |
| Syn (0.5, 0.5) | 87.09 | 86.35  | 86.98    | **89.07** |
| Syn (1.0, 1.0) | 90.52 | 89.37  | 90.12    | **91.70** |

The new ablation study reveals a significant performance disparity between weight assignments generated through optimal transport (MosT) and those generated randomly (`w-MGDA`, or MosT w/o OT). **This demonstrates the importance of optimal transport to achieve a balanced matching between solutions and objectives.**


***#4.3 Linearization with weights obtained from OT (i.e., effectiveness of MGDA in MosT)***

In the table above, we compared MosT against the variant `MosT (L)`, which uses linearization (L) (as opposed to MGDA) for MOO, with objective weights still generated by optimal transport. The details of this ablation study are presented in Appendix C.3. The experimental results showcase the consistent superiority of MosT over `MosT (L)` (or MosT w/o MGDA) on three federated learning datasets. **It proves the effectiveness of multi-objective optimization via MGDA for parameter updates.**

Sincerely,
Authors of Paper #8707

---

### Author Response · Authors · 2023-11-18
**General Response (part 1)**

We thank all reviewers for taking the time to review our work and for their constructive feedback. In the following, we respond to shared concerns and provide more experimental results.

---

**#1 Novelty and significance:**


MosT tackles a long-overlooked, underexplored, and critical challenge in many machine learning problems: optimizing multiple or **many (hundreds to thousands) objectives with a small set (tens) of models or solutions on the Pareto front**. This critical gap, previously underestimated, is pivotal in achieving optimal and balanced trade-offs among many objectives. By introducing a bi-level multi-objective optimization framework guided by optimal transport between objectives and models/solutions, MosT not only recognizes but decisively bridges this gap. MosT showcases its power and advantages over existing approaches in broad applications across federated learning, fairness-accuracy trade-offs, and multi-task learning.

**#2 Theoretical Foundations for Diversity:**

This paper takes the first step towards studying the multi-objective multi-solution transport problem. Our primary focus in this paper is to introduce a novel problem formulation, develop a practically effective algorithm with basic convergence guarantee, and showcase a wide range of practical applications of the formulation and algorithm with promising advantages over existing methods. That being said, we agree on the importance of further theoretical analysis about the diversity and will investigate it in future iterations of the work. The following is a more detailed discussion on the diversity of the solutions.

The optimal transport between solutions/models and objectives, specifically the two marginal constraints, plays a crucial role in achieving a diverse set of $m$ solutions with different yet complementary assignments or coverage on the $n$ objectives. For example, a uniform marginal over $n$ objectives enforces the $m$ models together cover all the $n$ objectives, and a uniform marginal over $m$ models avoids the cases when some models cover more objectives than others. Together, they enforce each model to cover a relatively different subset of objectives from other models, leading to the diversity. When $n$ is small, we can always augment the objectives as in Section 3.1 to secure the diversity. Exception happens when every model is large and powerful enough to cover all the objectives by itself, which is less interesting because in this case it is not necessary to train multiple models.


**#3 Time Complexity of OT and MGDA:**

OT and MGDA take up a remarkably small fraction of the total running time: for example, 0.06% for OT and 3.57% for MGDA (the min-norm optimization part) on average across all datasets in the experiments.

Moreover, when compared to baselines such as linearization and EPO, MosT remains more efficient overall.
In the table below, we report the end-to-end running time (seconds) of different approaches on two real datasets, DomainNet and Office-10. Particularly, we compare MosT's running time with two common baselines, i.e., MGDA and Linearization, and one baseline of interest, EPO.

|           | MGDA (s) | Linearization (s) | EPO (s) | MosT (s) |
|:---------:|:--------:|:-----------------:|:-------:|:--------:|
| DomainNet |  2109.28 |      3495.26      | 4220.37 |  2305.39 |
| Office-10 |  883.39  |      1726.79      | 2060.39 |  979.70  |

---

### Meta-Review · Area_Chair_4pRM · 2023-12-07

**Metareview:**

The reviewers feel that the paper contains interesting ideas but is lacking in proving or demonstrating the their benefits. In particular, the theoretical analyses only focus on approaching the Pareto set, but not on diversity of the solutions. The experimental results about a large set of objectives only contain quantitative measures on average accuracy, instead of diversity measures such as hypervolume (There is a figure of the KL divergence evolution. But the question is what's the rate of the diversity decay?). To be able to use the experimental results to substitute theoretical analysis, the authors would also need to compare with the state-of-the-art methods in empirical studies, such as some of the evolutionary computation methods.

**Justification For Why Not Higher Score:**

Perhaps the easiest way to improve the paper is to perform theoretical analysis on the diversity of the solutions. That would address many reviewers' concerns.

**Justification For Why Not Lower Score:**

n/a

---

### Decision · Program_Chairs · 2024-01-16

Reject